# FAST: Foreground-aware Diffusion with Accelerated Sampling Trajectory for Segmentation-oriented Anomaly Synthesis

Xichen Xu[1]    Yanshu Wang[1]    Jinbao Wang[2]    Xiaoning Lei[3]
Guoyang Xie[3*]    Guannan Jiang[3*]    Zhichao Lu[4]
[1]Global Institute of Future Technology, Shanghai Jiao Tong University, Shanghai, China
[2]School of Artificial Intelligence, Shenzhen University, Shenzhen, China
[3]Department of Intelligent Manufacturing, CATL, Ningde, China
[4]Department of Computer Science, City University of Hong Kong, Hong Kong, China
neptune_2333@sjtu.edu.cn   isaac_wang@sjtu.edu.cn   wangjb@szu.edu.cn
leixn01@catl.com  jianggn@catl.com  guoyang.xie@ieee.org   zhichao.lu@cityu.edu.hk

## Abstract

Industrial anomaly segmentation relies heavily on pixel-level annotations, yet real-world anomalies are often scarce, diverse, and costly to label. Segmentation-oriented industrial anomaly synthesis (SIAS) has emerged as a promising alternative; however, existing methods struggle to balance sampling efficiency and generation quality. Moreover, most approaches treat all spatial regions uniformly, overlooking the distinct statistical differences between anomaly and background areas. This uniform treatment hinders the synthesis of controllable, structure-specific anomalies tailored for segmentation tasks. In this paper, we propose FAST, a foreground-aware diffusion framework featuring two novel modules: the Anomaly-Informed Accelerated Sampling (AIAS) and the Foreground-Aware Reconstruction Module (FARM). AIAS is a training-free sampling algorithm specifically designed for segmentation-oriented industrial anomaly synthesis, which accelerates the reverse process through coarse-to-fine aggregation and enables the synthesis of state-of-the-art segmentation-oriented anomalies in as few as 10 steps. Meanwhile, FARM adaptively adjusts the anomaly-aware noise within the masked foreground regions at each sampling step, preserving localized anomaly signals throughout the denoising trajectory. Extensive experiments on multiple industrial benchmarks demonstrate that FAST consistently outperforms existing anomaly synthesis methods in downstream segmentation tasks. We release the code in https://github.com/Chhro123/fast-foreground-aware-anomaly-synthesis.

## 1 Introduction

**Motivation.** Industrial anomaly segmentation plays a vital role in modern manufacturing, aiming to localize abnormal regions at the pixel level. Unlike traditional anomaly detection, which typically performs binary classification at the image or region level, anomaly segmentation requires more fine-grained and precise localization of abnormal patterns. However, real-world anomalies are inherently scarce, diverse, and non-repeatable, making it difficult to collect data that fully captures the range of possible abnormal types. Moreover, acquiring high-quality pixel-level annotations is labor-intensive and costly, especially in industrial scenarios. To address these limitations, recent studies have increasingly explored the use of synthetic anomalies to expand the training data space and improve downstream performance.

---

[*]Corresponding authors.

39th Conference on Neural Information Processing Systems (NeurIPS 2025).

**Limitations.** Despite recent advances, current anomaly synthesis methods face three fundamental limitations that hinder their effectiveness for segmentation tasks [35]. (i) Lack of controllability. Most existing methods provide limited control over the structure, location, or extent of synthesized anomalies. This limitation is particularly evident in GAN-based approaches [22, 41, 6]. These methods typically adopt a one-shot generation paradigm, offering little flexibility in specifying where and how anomalies should appear. (ii) Neglect of segmentation-relevant properties. Training-free methods such as patch replacement or texture corruption [16, 40] may produce visible anomalies, but the synthesized patterns often lack the structural consistency and complexity of real-world industrial anomalies, which are critical for improving segmentation performance. (iii) Uniform treatment of spatial regions and inefficiency. Although recent diffusion-based methods [11, 14, 26] have mitigated the above issues, they still treat all spatial regions uniformly during both forward and reverse processes, without explicitly modeling the distinct statistical properties of anomaly regions [43, 38]. This absence of region-aware modeling prevents the model from preserving abnormal regions throughout the synthesis trajectory. Moreover, these models typically require hundreds to thousands of denoising steps [12, 27], resulting in a significant computational cost, especially for the real-world production line changeover. While recent training-free methods [17] aim to accelerate sampling, they fail to incorporate anomaly-aware cues, making them less effective for segmentation-oriented industrial anomaly synthesis (SIAS). These limitations motivate the need for SIAS models that support controllable anomaly synthesis, explicit modeling of anomaly regions, and efficient, task-aligned sampling strategies.

**FAST.** To address these issues, we propose FAST, a novel foreground-aware diffusion framework with two complementary modules: Anomaly-Informed Accelerated Sampling (AIAS) and the Foreground-Aware Reconstruction Module (FARM). (i) AIAS is a training-free sampling strategy that reduces the number of denoising steps by up to 99% (from 1000 to as few as 10), resulting in over 100× speedup for SIAS tasks. Despite this drastic acceleration, FAST achieves an average mIoU of 76.72% and accuracy of 83.97% on MVTec-AD, outperforming all prior state-of-the-art methods. (ii) FARM explicitly models abnormal regions by reconstructing pseudo-clean anomalies and generating anomaly-aware noise at each step in both the forward and reverse processes. Incorporating FARM boosts performance from 65.33% to 76.72% in mIoU (↑11.39), and from 71.24% to 83.97% in accuracy (↑12.73), demonstrating its critical role in enhancing anomaly salience. Detailed results are provided in Sec. 4.3. Together, AIAS and FARM enable FAST to generate controllable and segmentation-aligned anomalies that significantly improve downstream performance.

**Contributions.** In summary, our contributions are three-fold: (1) To mitigate the inefficiency and semantic misalignment of existing diffusion sampling, we introduce a training-free Anomaly-Informed Accelerated Sampling (AIAS) strategy that aggregates multiple denoising steps into a small number of coarse-to-fine analytical updates. (2) To address the lack of persistent anomaly-region representation, we propose a Foreground-Aware Reconstruction Module (FARM) that reconstructs pseudo-clean anomalies and reintegrates anomaly-aware noise at each step. (3) To support segmentation-oriented industrial anomaly synthesis, we design FAST, a controllable and efficient model. Extensive experiments on MVTec-AD and BTAD datasets demonstrate that it significantly outperforms existing methods in downstream segmentation tasks.

## 2 Related work

**Industrial Anomaly Synthesis.** Industrial anomaly synthesis aims to mitigate the scarcity of labeled abnormal samples in real-world inspection scenarios. Existing methods can be categorized into hand-crafted and DL-based approaches. Hand-crafted methods typically apply training-free manipulations to normal images, such as patch pasting [23, 25] or external texture blending [40, 36, 44] from sources like DTD [4], but they suffer from distributional deviation and limited realism. Deep learning-based methods alleviate these limitations by learning from real anomaly patterns. GAN-based methods [7, 32] can synthesize visually realistic anomalies but lack fine-grained controllability over anomaly shape and location. Diffusion-based methods [7, 15, 37, 10] offer stronger generative capacity via large-scale pretrained models, yet treat all regions uniformly and lack explicit control over anomaly localization, which is essential for segmentation. To this end, we propose FAST, which integrates foreground-aware reconstruction and efficient, segmentation-oriented anomaly synthesis into a unified diffusion framework.

**Acceleration of Discrete-Time Diffusion Models.** Diffusion models can be categorized into continuous-time and discrete-time frameworks. Continuous formulations [18, 19, 46] adopt SDE/ODE-based parameterizations and leverage high-order solvers for efficient sampling. In contrast,

standard DDPMs [12] model a discrete-time Markov chain with fixed variance schedules and require thousands of iterative denoising steps. While continuous-time solvers achieve notable speedups, they rely on continuously parameterized noise or score functions, which requires reformulating training objectives or interface in discrete-time models. Therefore, various acceleration techniques have been developed specifically for discrete-time diffusion. Some methods modify the generative process to reduce steps: DDGAN [33] integrates GAN-based decoding, TLDM [45] and ES-DDPM [20] truncate the forward process, and Blurring Diffusion Models [13] operate in the frequency domain. However, these methods require retraining and show limited generalization. In contrast, training-free approaches such as DDIM [27], PLMS [17], and GGDM [30] accelerate sampling without model modification. Yet, they treat all spatial regions uniformly and lack task-specific guidance essential for SIAS. Recent work like CUT [28] introduces external prompts for localized control for anomalies, but at the cost of multiple iterations per sampling step. In comparison, FAST proposes a novel training-free strategy that aggregates multiple denoising steps into coarse-to-fine segments while injecting mask-aware structural guidance, enabling efficient SIAS.

**Foreground–background Decoupling.** Foreground–background decoupling has been widely employed in industrial anomaly synthesis to enhance spatial precision and suppress irrelevant background interference. The core idea is to isolate defect-related regions from normal contexts, thereby improving downstream performance and synthesis controllability. Most methods such as PRN [42] and DCDGANc [31] perform explicit two-stage compositions, which first generate abnormal foregrounds and then blend them with normal backgrounds under soft mask constraints, but often suffer from boundary inconsistencies. Recent studies have introduced implicit separation; for instance, FCIS [29] enlarges the anomaly–background distance via contrastive learning, while BDG [3] incorporates masked attention and regularization within the denoiser to disentangle the influence of anomalies from the surrounding background. Although both BDG and FAST involve diffusion-based synthesis with certain forms of foreground–background decoupling, they pursue different research objectives through fundamentally distinct methodologies. FAST is a segmentation-oriented anomaly synthesis framework that emphasizes pixel-wise structural alignment and contextual consistency, whereas BDG primarily targets robust anomaly detection. Technically, AIAS in FAST analytically aggregates multiple DDPM reverse transitions into a few closed-form, coarse-to-fine updates, forming a deterministic and training-free sampler (e.g., $x_t \rightarrow x_{t-1}$) whose coefficients are precomputed under the original variance schedule, without any variance-controlling parameters like DDIM [27]. In contrast, BDG depends on DDIM inversion (e.g., $x_{t-1} \rightarrow x_t$) to maintain background features, which requires inversion consistency and retraining with regularization losses. These two mechanisms are fundamentally distinct and not directly interchangeable. Furthermore, the FARM module in FAST functions as an external foreground-reconstruction pathway that injects anomaly-aware noise via masks across timesteps to preserve anomaly salience throughout the sampling trajectory, whereas BDG employs masks merely as internal attention gates to localize edits within the denoiser. Essentially, BDG modifies the attention dynamics inside the denoiser to limit interference, while FARM operates outside the denoiser as a reconstruction-based enhancement module. These conceptual and algorithmic distinctions, together with different experiments and evaluation (segmentation-oriented mIoU/Acc vs. detection-oriented AUROC/AP) demonstrate that FAST and BDG follow independent research lines and remain technically and theoretically original.

# 3 Methods

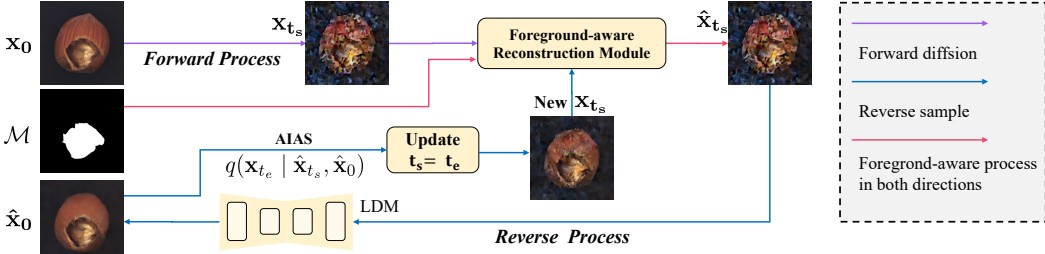

Figure 1: Illustration of a single forward–reverse process in FAST. AIAS accelerates sampling by aggregating multiple denoising steps into a small number of coarse-to-fine segments, achieving up to $100\times$ speedup while preserving semantic alignment under anomaly mask guidance. FARM extracts anomaly-only content from the noisy latent $x_t$ at each timestep $t$ and transforms it into anomaly-aware noise by re-applying forward diffusion.

**FAST for Anomaly Segmentation.** The proposed FAST framework is built upon the LDM [24] of $T$ steps. For notational simplicity, we denote the encoded latent of the original image as $x_0$, and its predicted reconstruction from the network as $\hat{x}_0$. We define $x_{t_s}$ as the noisy latent at timestep $t_s$, and $\hat{x}_{t_s}$ as the FARM-adjusted, anomaly-aware latent at the same step. Let $\mathcal{M} \in \{0,1\}^{H \times W}$ denote the binary anomaly mask, and $[t_s, t_e]$ represent a coarse-to-fine segment in AIAS, where $t_e < t_s$. Fig. 1 illustrates a single forward-reverse process at step $t_s$. In the forward phase, noise is added up to timestep $t_s$, yielding a noisy latent $x_{t_s}$. FARM ($F_\phi$ in Algorithm 1) then predicts a pseudo-clean anomaly latent $\hat{x}_0^{\text{an}}$, and adds noise to it up to timestep $t_s$ to obtain an anomaly-aware latent $\hat{x}_{t_s}$, which aims to match the observed $x_{t_s}$ in masked regions during training. In the corresponding reverse process, we divide the full denoising process into $S$ segments, each spanning $[t_s, t_e]$. Within each segment, AIAS approximates the posterior transition using: $q(x_{t_e} \mid x_{t_s}, \hat{x}_0)$. This formulation aggregates multiple DDPM steps into a single numerical update. FARM is also applied to refine $x_{t_e}$, ensuring the preservation of anomaly cues throughout the reverse process. More details can be seen in Algorithms 1 and 2. In addition, for the textual conditioning component of LDM, we follow the configuration of Anomaly Diffusion [15]; more implementation details can be found there.

---

**Algorithm 1** FAST Training

1: **repeat**
2:      $x_0 \sim q(x_0)$, $\mathcal{M}$, and weights $\lambda_1, \lambda_2$
3:      $t_s \sim \text{Uniform}(\{1, \ldots, T\})$, $\epsilon \sim \mathcal{N}(0, \mathbf{I})$
4:      $x_{t_s} = \sqrt{\bar{\alpha}_{t_s}} x_0 + \sqrt{1 - \bar{\alpha}_{t_s}} \epsilon$
5:      $\hat{x}_{t_s} = \sqrt{\bar{\alpha}_{t_s}} F_\phi(x_{t_s}, \mathcal{M}) + \sqrt{1 - \bar{\alpha}_{t_s}} \epsilon$
6:      Take gradient descent step on:

$$\nabla_\theta \| \epsilon - \epsilon_\theta(\hat{x}_{t_s}, t_s) \|^2$$

$$+ \nabla_\phi \| (\mathcal{M} \odot x_0 - F_\phi(x_{t_s}, t_s, \mathcal{M})) \|^2$$

7: **until** converged

**Algorithm 2** FAST Sampling

(Details are shown in Supplementary Material A.4)

1: Initialize $x_T \sim \mathcal{N}(0, \mathbf{I})$
2: **for** each segment $[t_s, t_e]$ from $T \to 0$ **do**
3:      $\hat{\epsilon} = \epsilon_\theta(x_{t_s}, t_s)$
4:      $\hat{x}_0 = \frac{1}{\sqrt{\bar{\alpha}_{t_s}}} (x_{t_s} - \sqrt{1 - \bar{\alpha}_{t_s}} \cdot \hat{\epsilon})$
5:      AIAS:

$$\mathbf{x}_{t_e} = F_\phi(q(x_{t_e} \mid x_{t_s}, \hat{x}_0), t_e, \mathcal{M}))$$

6: **end for**
7: **return** $x_0$

---

### 3.1 Anomaly-Informed Accelerated Sampling

The standard DDPM allows us to directly compute the marginal distribution of $x_t$ given a clean sample $x_0$ and additive noise $\epsilon$. Therefore, the one-step posterior distribution of $x_{t-1}$ can be expressed as:

$$q(x_{t-1} \mid x_t, x_0) = \mathcal{N}(A_t x_0 + B_t x_t, \sigma_t^2 \mathbf{I}), \tag{1}$$

where the coefficients are derived from the variance schedule as follows:

$$A_t = \frac{\sqrt{\bar{\alpha}_{t-1}} \beta_t}{1 - \bar{\alpha}_t}, \quad B_t = \frac{\sqrt{\alpha_t}(1 - \bar{\alpha}_{t-1})}{1 - \bar{\alpha}_t}, \quad \sigma_t^2 = \frac{1 - \bar{\alpha}_{t-1}}{1 - \bar{\alpha}_t} \beta_t,$$

and $\alpha_t = 1 - \beta_t$, $\bar{\alpha}_t = \prod_{s=1}^{t} \alpha_s$. All are closed-form coefficients derived from a predefined noise schedule. In practice, the true sample $x_0$ is not accessible during inference, and is typically replaced by a model prediction $\hat{x}_0$ obtained via denoising estimation. Equation 1 thus serves as the foundation for approximate posterior sampling, provided that $\hat{x}_0$ is a sufficiently accurate estimate of the ground truth $x_0$.

Theoretically, if we assume $\hat{x}_0 = x_0$ holds exactly (i.e., the prediction perfectly matches the ground-truth image), then the entire reverse process becomes fully deterministic and analytically tractable, with the only source of stochasticity being the injected noise at each step. In this idealized setting, the reverse sampling trajectory is fully governed by closed-form probabilistic transitions. This forms the basis for Lemma. 1 (For brevity, the full proof is provided in the Supplementary Material A.1).

**Lemma 1** (**Linear–Gaussian closure**). *Let $\{x_k\}_{k=0}^{K} \subset \mathbb{R}^d$ satisfy the recursion*

$$x_{k-1} = C_k x_k + d_k + \varepsilon_k, \quad \varepsilon_k \sim \mathcal{N}(0, \Sigma_k), \quad \varepsilon_k \perp \{x_k, \varepsilon_{k+1}, \ldots\}, \tag{2}$$

*where $C_k \in \mathbb{R}^{d \times d}$, $d_k \in \mathbb{R}^d$, and $\Sigma_k \in \mathbb{R}^{d \times d}$ are deterministic. Then, for every integer $m$ with $1 \leq m \leq k$, $x_{k-m}$ is again an affine–Gaussian function of $x_k$:*

$$x_{k-m} = \underbrace{\Big( \prod_{i=0}^{m-1} C_{k-i} \Big)}_{=:C_k^{(m)}} x_k + \underbrace{\sum_{i=0}^{m-1} \Big( \prod_{j=1}^{i} C_{k-j} \Big) d_{k-i}}_{=:d_k^{(m)}} + \varepsilon_k^{(m)}, \tag{3}$$

*where*

$$\varepsilon_k^{(m)} \sim \mathcal{N}\big(0, \Sigma_k^{(m)}\big), \qquad \Sigma_k^{(m)} = \sum_{i=0}^{m-1} \Big(\prod_{j=1}^{i} C_{k-j}\Big) \Sigma_{k-i} \Big(\prod_{j=1}^{i} C_{k-j}\Big)^\top.$$

While the ideal condition $\hat{x}_0 = x_0$ rarely holds in practice, the following properties justify the use of $\hat{x}_0$ in the multi-step formulation:

(i) The training objective of standard DDPM is explicitly designed to minimize the discrepancy between the predicted noise and the true noise. Consequently, the denoising model $\epsilon_\theta(x_t, t)$ implicitly learns to reconstruct a close approximation of $x_0$ through the reverse reparameterization formula.

(ii) Both empirical observations and theoretical analyses suggest that $\hat{x}_0$ varies slowly with respect to $t$ at large diffusion steps. That is, for a segment $[t_s, t_e]$ with $t_s > t_e$ and moderate length (e.g., $t_s - t_e \ll T$), we have $\hat{x}_0(x_{t_s}, t_s) \approx \hat{x}_0(x_t, t)$ for all $t \in [t_s, t_e]$, due to the temporal smoothness of model predictions in the noise-dominated regime.

Therefore, it is reasonable to treat $\hat{x}_0$ as fixed within a short temporal window. Under this assumption, multiple single-step reverse transitions can be analytically composed into a single multi-step affine–Gaussian kernel. This approximation and Lemma 1 form the basis for Theorem 2, which characterizes the closed-form reverse process from $t_s$ to $t_e$ (For brevity, the full proof is provided in the Supplementary Material A.2 ).

**Lemma 2** (Closed-form reverse from $\boldsymbol{t}_s \to t_e$). *Fix indices $0 \le t_e < t_s \le T$, and let the single-step coefficients $(A_t, B_t, \sigma_t^2)$ be defined as in Eq. 12. Then the aggregated reverse kernel over $t_s \to \cdots \to t_e$ is affine–Gaussian:*

$$x_{t_e} = \Pi_{t_e}^{t_s} x_{t_s} + \Sigma_{t_e}^{t_s} \hat{x}_0 + \varepsilon_{t_e}, \tag{4}$$

*where*

$$\Pi_{t_e}^{t_s} := \prod_{i=t_e+1}^{t_s} B_i, \quad \Sigma_{t_e}^{t_s} := \sum_{i=t_e+1}^{t_s} A_i \prod_{j=i+1}^{t_s} B_j, \quad \varepsilon_{t_e} \sim \mathcal{N}\left(0, \sum_{i=t_e+1}^{t_s} \left(\prod_{j=i+1}^{t_s} B_j\right)^2 \sigma_i^2 \mathbf{I}\right).$$

Therefore, it can be observed that in the limited segments (e.g., $t_s \to t_e$), there are the three scalars $\big(\Pi_{t_e}^{t_s}, \Sigma_{t_e}^{t_s}, \varepsilon_{t_e}\big)$, allowing us to precompute them once and re-use them during sampling. Lemma. 2 enables theoretical computation of posterior transitions between any two timesteps $t_s$ and $t_e$, allowing multi-step sampling in a manner distinct from DDIM. However, while the affine–Gaussian transition provides an efficient coarse approximation for the reverse path $x_{t_s} \to x_{t_e}$, the approximation may introduce residual artifacts in parctice. It is caused by the strong noise attenuation and the fixed $\hat{x}_0$ assumption. Moreover, since $x_t$ inherently entangles both the foreground and the background content, direct sampling through the affine-Gaussian kernel will ignore the critical spatial structure discrepancies for SIAS.

To better preserve anomaly-localized information while ensuring smooth global composition, we explicitly decompose the clean sample $x_0$ into two disjoint components:

$$\mathbf{x}_0 = \mathbf{x}_0^{\text{an}} + \mathbf{x}_0^{\text{bg}}, \tag{5}$$

where $\mathbf{x}_0^{\text{an}}$ is the anomaly-only region (masked by $\mathcal{M}$), and $\mathbf{x}_0^{\text{bg}}$ is the background. The background is independently forward-diffused:

$$\mathbf{x}_{t_e}^{\text{bg}} \sim q(\mathbf{x}_{t_e}^{\text{bg}} \mid \mathbf{x}_0^{\text{bg}}), \tag{6}$$

while the anomaly foreground is refined by the learned FARM module (introduced later in Sec. 3.2), and merged with the background through spatial masking:

$$\mathbf{x}_{t_e}^R = \text{FARM}(\mathbf{x}_{t_e}), \qquad \mathbf{x}_{t_e} = \mathcal{M} \odot \mathbf{x}_{t_e}^R + (1 - \mathcal{M}) \odot \mathbf{x}_{t_e}^{\text{bg}}. \tag{7}$$

This foreground-aware fusion ensures consistent noise levels between anomalous and normal regions at each step, preserving local anomaly salience while maintaining global visual coherence. In practice, we also introduce a final fine-grained refinement stage using standard DDPM posterior sampling for small $t$ (e.g., $t = 1$ or $t = 2$) to restore the alignment between the coarse trajectory and the ground-truth posterior, and to enhance fine-scale texture fidelity. The complete sampling algorithm is summarized in Algorithm 3.

## 3.2 Foreground-Aware Reconstruction Module

As discussed above, conventional diffusion models treat all spatial regions uniformly, which limits their ability to synthesize localized anomalies. To address this, we propose the Foreground-Aware Reconstruction Module (FARM), which reconstructs clean anomaly-only content from noisy latent inputs under both temporal and spatial guidance. As illustrated in Fig. 2, FARM adopts an encoder–decoder architecture. The encoder $f_{\text{enc}}$ extracts deep representations from the noisy latent $x_{t_s}$, while the decoder $f_{\text{dec}}$ progressively upsamples and integrates the binary mask $\mathcal{M}$ at multiple resolutions, ensuring spatial alignment with anomaly regions throughout the hierarchy.

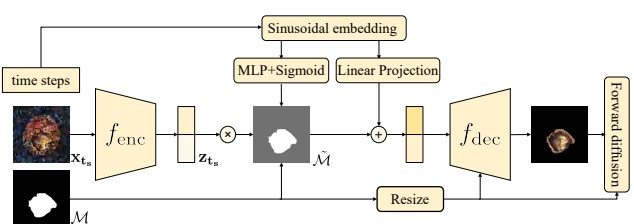

Figure 2: The architecture of FARM. Given noisy latent $x_{t_s}$ and mask $\mathcal{M}$, the encoder $f_{\text{enc}}$ extracts features $z_{t_s}$, which is also modulated by a background-adaptive soft mask $\tilde{\mathcal{M}}$ and related timestep embedding $\boldsymbol{\tau}_{t_s}$. The decoder $f_{\text{dec}}$ then reconstructs the anomaly-only latent $\hat{x}_0^{\text{an}}$, which is forward-diffused to produce anomaly-aware noise.

To encode temporal context, we initialize sinusoidal timestep embeddings $\boldsymbol{\tau}_{t_s} \in \mathbb{R}^d$ and project them into latent space via a learned linear layer. These embeddings are added to the encoder output, modulating feature responses based on the current noise level and allowing the decoder to reconstruct temporally consistent structures.

In addition, to modulate background activation, we introduce a background-adaptive soft mask:

$$\tilde{\mathcal{M}} = \mathcal{M}_d + (1 - \mathcal{M}_d) \cdot \sigma(f_{\text{bg}}(\boldsymbol{\tau}_{t_s})), \tag{8}$$

where $\mathcal{M}_d$ is a downsampled binary mask aligned with encoder resolution, and $f_{\text{bg}}$ is a lightweight MLP. This design allows FARM to suppress irrelevant background features while adapting to the current timestep.

The encoded feature is computed as:

$$z_{t_s} = \tilde{\mathcal{M}} \cdot f_{\text{enc}}(\mathbf{x}_{t_s}) + \text{Proj}(\boldsymbol{\tau}_{t_s}), \tag{9}$$

and decoded into an anomaly-only latent: $\hat{x}_0^{\text{an}} = f_{\text{dec}}(z_{t_s}, \mathcal{M})$.

To inject anomaly-aware noise into the sampling trajectory, the reconstructed anomaly is forward-diffused:

$$\hat{x}_{t_s}^{\text{an}} = \sqrt{\bar{\alpha}_{t_s}} \cdot \hat{x}_0^{\text{an}} + \sqrt{1 - \bar{\alpha}_{t_s}} \cdot \epsilon, \quad \epsilon \sim \mathcal{N}(0, \mathbf{I}), \tag{10}$$

and replaces the original noise in masked regions:

$$\hat{x}_{t_s} = (1 - \mathcal{M}) \cdot x_{t_s} + \mathcal{M} \cdot \hat{x}_{t_s}^{\text{an}}. \tag{11}$$

During training, FARM is supervised to ensure that the reconstructed anomalies match the masked regions of the noisy inputs. During inference, temporal and spatial guidance together enable FARM to introduce localized and temporally coherent anomaly signals into the reverse trajectory, ensuring alignment with the global generative process while enhancing fine-grained control.

## 4 Experiments

### 4.1 Implementation Details.

**Datasets.** We evaluate FAST on two widely-used industrial anomaly segmentation benchmarks: MVTec-AD [1] and BTAD [21]. For each anomaly class, we synthesize image–mask pairs using normal images, binary masks, and text prompts describing anomaly semantics. A total of 500 samples are generated for each anomaly type within a class, with approximately one-third used for training and the remainder reserved for evaluation. This design ensures sufficient structural diversity while maintaining training efficiency. **Mask Generation Strategy.** Our mask synthesis consists of two complementary components: (i) geometric augmentation of real anomaly masks via operations like rotation and flipping; (ii) synthesis of new masks using a Latent Diffusion Model (LDM) pre-trained on real anomaly mask examples, which follows the protocol of AnomalyDiffusion [15]. All

synthesized masks undergo manual screening to guarantee visual realism, structural diversity, and consistency with typical industrial abnormal structures. **Evaluation Metrics.** We report performance using mean intersection over union (mIoU) and pixel-wise accuracy (Acc), following standard practice in anomaly segmentation. **Baselines.** FAST is compared against six representative anomaly synthesis approaches: CutPaste [16], DRAEM [40], GLASS [2], the GAN-based SOTA method DFMGAN [8], and diffusion-based SOTA models Anomaly Diffusion [15] and RealNet [43]. To simulate realistic deployment scenarios, we pair all generation methods with lightweight segmentation networks, including Segformer [34], BiSeNet V2 [39], and STDC [9]. As our method adopts the same prompt-driven synthesis setup as AnomalyDiffusion [15], we omit the details here for brevity. Full specifications of the textual configuration, as well as other implementation details, including dataset preprocessing, sampling schedules, loss weights, and hyperparameter settings, are provided in the Supplementary Materials A.5.

## 4.2   Comparison Studies

Table 1: Evaluation of pixel-level segmentation accuracy on extended MVTec data using real-time Segformer. Detailed per-category results for other real-time segmentation model, such as BiseNet V2 and STDC are reported in Supplementary Material A.6.

| Category | CutPaste | | DRAEM | | GLASS | | DFMGAN | | RealNet | | AnomalyDiffusion | | FAST | |
|---|---|---|---|---|---|---|---|---|---|---|---|---|---|---|
| | mIoU ↑ | Acc ↑ | mIoU ↑ | Acc ↑ | mIoU ↑ | Acc ↑ | mIoU ↑ | Acc ↑ | mIoU ↑ | Acc ↑ | mIoU ↑ | Acc ↑ | mIoU ↑ | Acc ↑ |
| bottle | 75.11 | 79.49 | 79.51 | 84.99 | 70.26 | 76.30 | 75.45 | 80.39 | 77.96 | 83.90 | 76.39 | 83.54 | 86.86 | 90.90 |
| cable | 55.40 | 60.49 | 64.52 | 70.77 | 58.81 | 62.32 | 62.10 | 64.87 | 62.51 | 69.27 | 62.49 | 74.48 | 73.71 | 77.94 |
| capsule | 35.15 | 40.29 | 51.39 | 62.32 | 34.12 | 38.04 | 41.29 | 45.83 | 46.76 | 51.91 | 37.73 | 44.72 | 63.22 | 71.12 |
| carpet | 66.34 | 77.59 | 72.57 | 81.28 | 70.11 | 77.56 | 71.33 | 83.69 | 68.84 | 79.15 | 64.67 | 73.59 | 73.84 | 83.53 |
| grid | 29.90 | 46.72 | 47.75 | 67.85 | 37.43 | 46.30 | 37.73 | 54.13 | 37.55 | 48.86 | 38.70 | 51.82 | 52.45 | 70.70 |
| hazel_nut | 56.95 | 60.72 | 84.22 | 89.74 | 55.51 | 57.43 | 83.43 | 86.03 | 60.18 | 63.49 | 59.33 | 67.48 | 90.81 | 94.79 |
| leather | 57.23 | 63.49 | 64.12 | 71.49 | 62.05 | 73.38 | 60.96 | 68.02 | 68.29 | 77.16 | 56.45 | 62.51 | 66.60 | 74.18 |
| metal_nut | 88.78 | 90.94 | 93.51 | 96.10 | 88.15 | 90.52 | 92.77 | 94.93 | 91.28 | 94.09 | 88.00 | 91.10 | 94.65 | 96.88 |
| pill | 43.28 | 47.11 | 46.99 | 49.76 | 41.52 | 43.54 | 87.19 | 90.05 | 47.32 | 58.31 | 83.21 | 89.00 | 90.17 | 94.07 |
| screw | 25.10 | 31.35 | 46.96 | 59.03 | 35.94 | 42.37 | 46.65 | 50.79 | 47.12 | 55.17 | 38.47 | 49.49 | 49.94 | 57.48 |
| tile | 85.33 | 91.60 | 89.21 | 93.74 | 85.67 | 90.28 | 88.87 | 91.96 | 83.53 | 87.30 | 84.29 | 89.72 | 90.13 | 93.77 |
| toothbrush | 39.40 | 63.93 | 65.35 | 79.43 | 53.75 | 60.46 | 61.00 | 70.50 | 57.68 | 72.03 | 48.68 | 64.41 | 74.98 | 88.63 |
| transistor | 65.03 | 71.05 | 59.96 | 62.18 | 29.28 | 30.67 | 73.56 | 78.48 | 63.71 | 66.79 | 79.27 | 91.74 | 91.80 | 94.50 |
| wood | 49.64 | 60.47 | 67.52 | 73.28 | 50.91 | 53.16 | 67.00 | 80.84 | 61.84 | 89.54 | 60.16 | 74.62 | 78.77 | 86.31 |
| zipper | 65.39 | 71.89 | 69.29 | 79.36 | 69.98 | 79.31 | 66.34 | 70.50 | 68.78 | 78.50 | 65.36 | 72.66 | 72.80 | 84.73 |
| Average | 55.87 | 63.81 | 66.86 | 74.75 | 56.23 | 61.44 | 67.71 | 74.07 | 62.89 | 71.70 | 62.88 | 72.06 | 76.72 | 83.97 |

Table 2: Evaluation of pixel-level segmentation accuracy on extended BTAD data using real-time Segformer, BiseNet V2 and STDC.

| Backbone | Category | CutPaste | | DRAEM | | GLASS | | DFMGAN | | RealNet | | AnomalyDiffusion | | FAST | |
|---|---|---|---|---|---|---|---|---|---|---|---|---|---|---|---|
| | | mIoU ↑ | Acc ↑ | mIoU ↑ | Acc ↑ | mIoU ↑ | Acc ↑ | mIoU ↑ | Acc ↑ | mIoU ↑ | Acc ↑ | mIoU ↑ | Acc ↑ | mIoU ↑ | Acc ↑ |
| Segformer | 01 | 66.94 | 78.20 | 67.86 | 80.14 | 68.02 | 79.57 | 67.02 | 78.03 | 67.17 | 80.20 | 66.55 | 76.31 | 75.93 | 86.12 |
| | 02 | 65.04 | 83.64 | 69.52 | 82.96 | 69.99 | 83.58 | 68.75 | 84.92 | 70.64 | 83.90 | 68.06 | 84.74 | 70.63 | 81.63 |
| | 03 | 50.96 | 60.41 | 50.39 | 54.30 | 51.77 | 53.53 | 38.95 | 41.55 | 48.76 | 57.50 | 54.85 | 80.20 | 79.40 | 85.64 |
| BiseNet V2 | 01 | 57.15 | 69.88 | 49.16 | 65.48 | 44.09 | 50.57 | 49.49 | 59.20 | 45.45 | 57.65 | 46.66 | 55.18 | 58.74 | 68.98 |
| | 02 | 59.45 | 82.05 | 66.46 | 80.29 | 66.37 | 79.46 | 66.02 | 79.21 | 66.11 | 81.67 | 65.57 | 84.00 | 68.02 | 82.40 |
| | 03 | 31.84 | 40.62 | 36.15 | 39.04 | 30.80 | 37.15 | 20.12 | 21.48 | 29.55 | 33.11 | 42.27 | 74.41 | 77.87 | 92.49 |
| STDC | 01 | 48.06 | 59.86 | 42.17 | 65.36 | 45.51 | 60.12 | 44.68 | 51.71 | 32.91 | 49.21 | 44.85 | 55.29 | 44.95 | 53.47 |
| | 02 | 59.80 | 77.57 | 64.96 | 84.32 | 65.02 | 81.94 | 64.85 | 75.32 | 64.00 | 82.64 | 64.73 | 78.93 | 67.76 | 82.16 |
| | 03 | 19.76 | 25.20 | 36.14 | 38.80 | 17.04 | 28.01 | 14.67 | 16.55 | 22.57 | 24.79 | 41.71 | 65.45 | 84.04 | 92.36 |

**Anomaly Segmentation** Table. 1 and 2 report pixel-level segmentation results on various datasets using Segformer trained with FAST-augmented data. We observe that FAST achieves an average mIoU of 76.72% and accuracy of 83.97%, significantly outperforming the strongest prior method, DRAEM (74.75% Acc), by 9.22 points, respectively. Improvements are particularly notable in challenging categories: in *capsule*, FAST increases mIoU from 51.39% (DRAEM) to 63.22% (↑11.83); on *grid*, from 47.75% to 52.45% (↑4.70); and on *transistor*, from 84.22% to 91.80% (↑7.58). Even in relatively easier categories such as *bottle* and *tile*, FAST still yields consistent improvements of 7.35 and 0.92 mIoU points, respectively. These results demonstrate that the combination of mask-aware noise injection via FARM and coarse-to-fine accelerated sampling via AIES enables more realistic and structurally coherent anomaly synthesis, leading to superior segmentation performance. Similar trends are observed when replacing Segformer with other real-time backbones such as BiseNetV2 and STDC, as shown in Supplementary Materials A.6, confirming the generalizability of FAST across different segmentation architectures.

**Qualitative Comparison.** Fig. 3 visually compares anomaly samples synthesized by different anomaly synthesis methods across several MVTec-AD categories. It can be observed that traditional unsupervised methods such as CutPaste and DRAEM generate anomalies by overlaying arbitrary textures or patches without any semantic guidance. For instance, in the *cable* category, anomalies produced by CutPaste appear as artificial, block-like overlays lacking meaningful texture or structure. Similarly, DRAEM and GLASS introduce unrealistic color distortions and incoherent patterns in the *transistor* category, which deviate significantly

from typical industrial anomalies. DL-based approaches (DFMGAN, RealNet, and Anomaly-Diffusion) generate more visually plausible results, but still exhibit noticeable shortcomings.

For instance, RealNet often introduces color shifts and boundary artifacts, as seen in the *tile* and *cable* cases, where anomalies appear overly smooth or blurred. DFMGAN and AnomalyDiffusion are able to synthesize more coherent shapes (e.g., spray-paint-like anomalies in *hazel_nut*), yet they suffer from inaccurate boundaries and structural mismatches, as is especially evident in the *tile* (AnomalyDiffusion) and *cable* (DFMGAN) categories. In contrast, FAST consistently produces anomalies that closely resemble realistic anomalies while maintaining precise alignment with the annotated masks. In the *metal_nut* and *hazel_nut* cases, FAST is the only method that preserves fidelity and shape within the intended regions, demonstrating superior controllability and structural consistency. These results validate the effectiveness of the proposed FAST in segmentation-oriented anomaly synthesis.

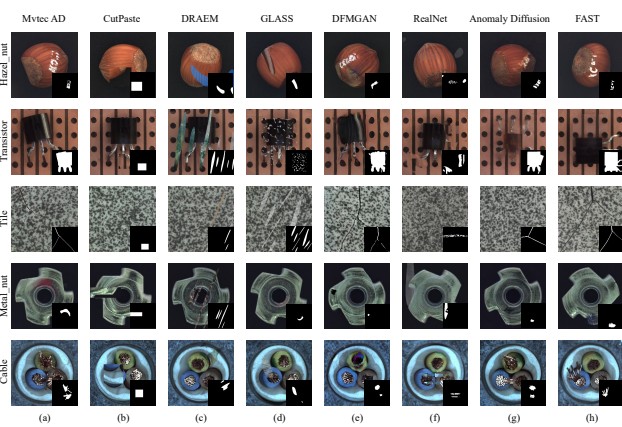

Figure 3: Visualization results of different anomaly synthesis methods on the MVTec dataset. Columns correspond to synthesis methods (from left to right: MVTec AD, CutPaste, DRAEM, GLASS, DFMGAN, RealNet, Anomaly Diffusion, FAST), and rows correspond to product categories (from top to bottom: hazel_nut, transistor, tile, metal_nut, cable).

### 4.3 Ablation Studies

**The Impact of AIAS.** We compare our proposed AIAS strategy with several widely-used training-free samplers, including DDPM [12] with 1000 steps, DDIM [27] with 50 steps and PLMS [17] with 50 steps. These methods represent state-of-the-art discrete-time sampling approaches for diffusion-based models. To ensure fairness, we exclude continuous-time solvers, as they rely on a fundamentally different formulation based on ODEs or SDEs, which necessitates a distinct training paradigm and architectural adjustments incompatible with our discrete-time framework. Quantitative results are reported in Fig. 5. While DDPM achieves competitive results on certain categories (e.g., *carpet*, *tile*), it requires 1000 iterative steps, making it over 20× slower than AIAS in practice.

DDIM and PLMS, though more efficient, exhibit inconsistent performance across categories and often underperform AIAS, particularly on challenging textures such as *capsule*, *grid*, and *transistor*. In contrast, AIAS achieves the best results on the majority of categories and consistently provides competitive or superior performance in both mIoU and accuracy, demonstrating its ability to generate segmentation-aligned anomalies with significantly fewer steps. It further indicates that by analytically aggregating multiple DDPM transitions into coarse-to-fine segments, AIAS reduces the discretization error inherent in single-step samplers (e.g., DDIM) or fixed multistep solvers (e.g., PLMS), allowing a closer approximation of the true posterior within just 50 steps. Fig. 4 further

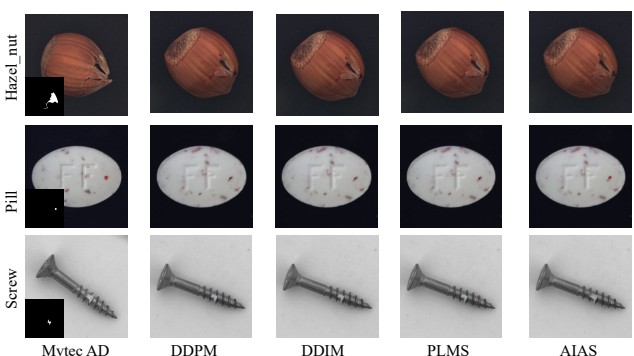

Figure 4: SIAS results with other sampling strategies. Columns correspond to sampling strategies (from left to right: ground truth, DDPM (1000 steps), DDIM (50 steps), PLMS (50 steps), AIAS (50 steps), and rows correspond to categories (from top to bottom: hazelnut, pill, screw). Further qualitative results (trained on MVTec and BTAD) are provided in the Supplementary Materials A.8.

illustrates the qualitative advantage. For example, in the *hazel_nut* class, the anomalies produced

by DDPM, DDIM, and PLMS display noticeable color inconsistencies near the anomaly boundary, resulting from distributional mismatch with the background. In comparison, FAST-produced anomalies that are well blended into the context, with sharper and more realistic structural alignment.

Although this result may seem counterintuitive, since fewer sampling steps usually imply degraded visual quality. And we believe the difference primarily stems from the evaluation objective. Specifically, DDPM sampling remains the best performer in terms of pure visual fidelity metrics in our work, but AIAS is designed to optimize downstream segmentation performance rather than perceptual realism alone. As shown in Table 3, moderately increasing the sampling steps can slightly enhance image quality, yet it also leads to a substantial rise in inference time. More importantly, excessive steps tend to weaken the anomaly localization consistency and thus degrade segmentation performance. Therefore, AIAS achieves a more favorable trade-off between SIAS and visual fidelity.

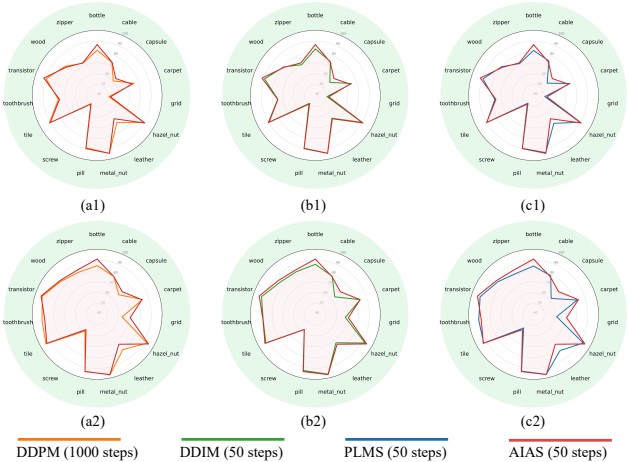

Figure 5: The effect of different sampling methods on SIAS in the MVTec dataset. **Top row** shows per-category segmentation performance using mIoU; **bottom row** shows performance using Acc. Detailed per-category results of AIAS are reported in Supplementary Material A.6.

Table 3: Comparison of pixel-level anomaly segmentation using different steps on the MVTec dataset.

| Category | Step 2 | | Step 5 | | Step 10 | | Step 30 | | Step 50 | | Step 100 | | Step 200 | | Step 500 | | Step 1000 | |
|---|---|---|---|---|---|---|---|---|---|---|---|---|---|---|---|---|---|---|
| | mIoU↑ | Acc↑ | mIoU↑ | Acc↑ | mIoU↑ | Acc↑ | mIoU↑ | Acc↑ | mIoU↑ | Acc↑ | mIoU↑ | Acc↑ | mIoU↑ | Acc↑ | mIoU↑ | Acc↑ | mIoU↑ | Acc↑ |
| bottle | 77.03 | 80.96 | 80.55 | 85.08 | 83.26 | 85.90 | 84.59 | 87.89 | **86.86** | **90.90** | 83.75 | 86.95 | 84.04 | 88.54 | 83.52 | 88.19 | 81.65 | 84.83 |
| cable | 47.39 | 48.66 | 69.58 | 73.11 | 71.23 | 75.07 | 73.34 | 77.59 | 73.71 | 77.94 | 72.99 | 76.50 | 72.83 | 76.51 | **75.23** | **79.32** | 73.45 | 78.06 |
| capsule | 43.56 | 48.58 | 49.81 | 54.22 | 54.85 | 59.31 | 61.12 | 67.08 | **63.22** | 71.12 | 63.15 | 71.17 | 62.12 | **71.76** | 62.83 | 70.88 | 60.01 | 66.87 |
| carpet | 70.24 | 80.98 | 73.22 | 83.18 | 73.10 | 84.06 | 73.56 | 80.50 | 73.84 | 83.53 | 73.41 | 82.92 | 73.17 | 81.90 | 73.27 | 82.49 | **75.99** | **84.14** |
| grid | 48.15 | 61.75 | 50.03 | 63.28 | 50.89 | 71.35 | 48.76 | 61.17 | **52.45** | **70.70** | 50.03 | 65.41 | 52.06 | 67.28 | 49.18 | 63.63 | 50.91 | 63.19 |
| hazel_nut | 76.16 | 78.75 | 84.16 | 86.50 | 90.45 | 94.04 | 90.49 | 94.04 | 90.81 | **94.79** | 90.82 | 94.16 | **90.87** | 94.27 | 90.77 | 94.71 | 89.81 | 93.31 |
| leather | 62.11 | 66.86 | 66.74 | 76.16 | **67.09** | **76.51** | 65.44 | 72.41 | 66.60 | 74.18 | 66.88 | 74.22 | 65.87 | 87.88 | 67.95 | 83.62 | 71.03 | 80.32 |
| metal_nut | 92.06 | 93.57 | 93.94 | 95.72 | 94.71 | 96.98 | 94.47 | 96.31 | 94.65 | 96.88 | **94.74** | **97.19** | 94.50 | 96.59 | 94.72 | 96.80 | 94.63 | 97.18 |
| pill | 50.03 | 55.46 | 80.01 | 82.53 | 90.07 | 93.80 | 90.02 | 94.24 | **90.17** | **94.07** | 89.82 | 94.10 | 89.80 | 93.22 | 90.15 | 94.34 | 89.36 | 93.79 |
| screw | 46.07 | 52.01 | 47.92 | 56.55 | 50.04 | 56.21 | **50.11** | **60.85** | 49.94 | 57.48 | 50.06 | 58.66 | 48.41 | 61.05 | 47.71 | 54.90 | 49.35 | 59.18 |
| tile | 87.26 | 93.92 | 89.46 | 94.96 | 89.72 | 93.92 | 89.58 | 93.68 | 90.13 | 93.77 | 89.93 | 94.45 | 90.02 | 93.73 | 89.71 | 93.38 | **91.01** | **94.72** |
| toothbrush | 58.54 | 67.15 | 76.65 | 87.41 | **76.96** | 90.29 | 74.36 | 90.78 | 74.98 | 88.63 | 74.17 | 87.29 | 73.32 | 86.49 | 75.66 | 89.50 | 76.10 | **91.25** |
| transistor | 66.42 | 71.59 | 66.08 | 70.23 | 77.27 | 79.66 | 89.45 | 92.65 | **91.80** | **94.50** | 91.39 | 94.66 | 89.67 | 93.50 | 90.32 | 93.21 | 89.59 | 93.41 |
| wood | 68.69 | 78.28 | 74.23 | 81.07 | 75.97 | 81.18 | 78.76 | 84.99 | 78.77 | **86.31** | 77.00 | 83.95 | 77.60 | 82.85 | 77.71 | 83.45 | **80.03** | 85.30 |
| zipper | 68.85 | 75.26 | 70.92 | 81.44 | 72.44 | 84.99 | **73.08** | 81.91 | 72.80 | **84.73** | 71.99 | 81.94 | 71.71 | 82.21 | 71.73 | 83.47 | 72.45 | 82.35 |
| Average | 64.17 | 70.25 | 71.55 | 78.10 | 74.54 | 81.55 | 75.81 | 82.41 | **76.72** | **83.97** | 76.01 | 82.90 | 75.73 | 83.85 | 76.03 | 83.46 | 76.36 | 83.19 |

Figure 6: Segmentation-oriented industrial anomaly synthesis results at different steps of AIAS. Columns correspond to increasing sampling steps $T$ (from left to right), and rows correspond to product categories (from top to bottom: capsule, bottle, carpet).

**The Impact of AIAS under different steps.** We further investigate the segmentation performance of AIAS under varying numbers of reverse steps, ranging from 2 to 1000, as reported in Table. 3. Remarkably, AIAS approximates the performance of full-step DDPM using only 10 steps, and reaches near-optimal results by 50 steps, demonstrating the effectiveness of our coarse-to-fine aggregation strategy. Performance improves rapidly as $t$ increases from 2 to 50, since early segments capture the global layout and coarse structure of anomalies, which are most relevant for segmentation. This trend is also visually confirmed in Fig. 6. Beyond this point, performance gains gradually saturate, indicating that additional steps primarily refine high-frequency details with limited impact on segmentation accuracy. Notably, when $t = 1000$, AISA degenerates to the original DDPM sampling

process, where each segment $[t_e, t_s]$ corresponds to a single denoising step. The convergence of performance at this point validates that our multi-step analytical updates provide a faithful approximation of the full diffusion trajectory, preserving both global semantics and fine-grained anomaly cues while significantly reducing sampling cost. Furthermore, excessive denoising steps may introduce over-smoothing or amplify reconstruction inconsistencies, potentially weakening the alignment between synthesized anomalies and segmentation-relevant structures. Overall, these results highlight that AIAS not only accelerates sampling, but also introduces an inductive structural bias beneficial for anomaly segmentation. In practice, the optimal balance between quality and efficiency is achieved within 10–50 steps.

**The Impact of FARM.** To evaluate the effectiveness of FARM, we conduct an ablation study by comparing the model's performance with (w/ FARM) and without (w/o FARM) FARM under identical AIAS settings. Results on the MVTec dataset are reported in Fig. 8. The inclusion of FARM leads to substantial improvements in segmentation performance, with average mIoU increasing from 65.33 to 76.42 and accuracy increasing from 71.24 to 83.97. The performance gains are particularly pronounced in challenging categories characterized by fine-grained or complex structures, such as *capsule* (↑14.1 mIoU), *grid* (↑14.7 mIoU), and

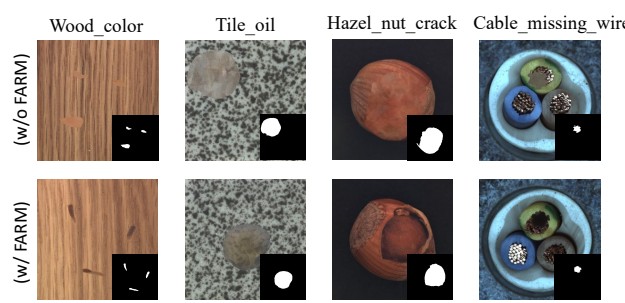

Figure 7: Qualitative ablation results with and without FARM on MVtec dataset. Columns correspond to category–anomaly pairs (from left to right: Wood_color, Tile_oil, Hazel_nut_crack, Cable_missing_wire; and rows correspond to ablation strategy (from top to bottom: without FARM (*w/o FARM*) and with FARM (*w/ FARM*).

*transistor* (↑29.5 mIoU). Even in relatively easier categories like *tile* and *hazel_nut*, FARM consistently enhances accuracy and boundary localization, as shown in Fig. 7. More detailed analysis of FARM can be found in Supplementary Material A.7.

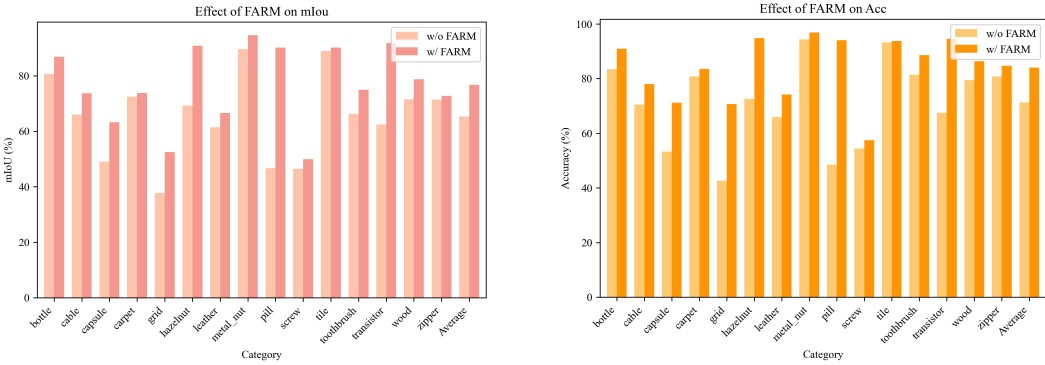

Figure 8: Qualitative ablation results with and without FARM on MVtec dataset. Columns correspond to product categories and rows correspond to mIou and Acc). Detailed per-category results for ablation study of FARM are reported in Supplementary Material A.7.

## 5   Conclusion

In this work, we proposed FAST, a segmentation-oriented foreground-aware diffusion framework tailored for anomaly synthesis. To address the limitations of existing anomaly synthesis methods, specifically their limited controllability and lack of structural awareness, we introduced two key components: the Foreground-Aware Reconstruction Module (FARM), which adaptively injects anomaly-aware noise at each sampling step, and the Anomaly-Informed Efficient Sampling (AIAS), a training-free strategy that accelerates sampling via coarse-to-fine aggregation. Built upon a discrete-time latent diffusion backbone, FAST enables the synthesis of segmentation-aligned anomalies with as few as 10 denoising steps. Extensive experiments on MVTec-AD and BTAD demonstrate that FAST outperforms existing baselines in downstream segmentation. FAST represents a promising step toward controllable and efficient segmentation-oriented industrial anomaly synthesis.

## Acknowledgments

This work was supported by the National Natural Science Foundation of China (Grant No. 62206122) and the Tencent "Rhinoceros Birds" — Scientific Research Foundation for Young Teachers of Shenzhen University.

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

# NeurIPS Paper Checklist

1. **Claims**

   Question: Do the main claims made in the abstract and introduction accurately reflect the paper's contributions and scope?

   Answer: [Yes]

   Justification: The abstract and introduction accurately reflect the contributions and scope of the paper. The paper introduces FAST, a foreground-aware diffusion framework with two core modules: Anomaly-Informed Accelerated Sampling (AIAS), which enables coarse-to-fine training-free sampling with up to 100× speed-up, and the Foreground-Aware Reconstruction Module (FARM), which constructs anomaly-aware noise at each denoising step to enhance abnormal regions. These claims are substantiated by theoretical derivations, algorithmic design, and comprehensive experiments showing consistent improvements on MVTec and BTAD datasets. The introduction does not overclaim or extend beyond the scope addressed in the experiments, and the focus remains tightly aligned with segmentation-oriented industrial anomaly synthesis.

   Guidelines:
   - The answer NA means that the abstract and introduction do not include the claims made in the paper.
   - The abstract and/or introduction should clearly state the claims made, including the contributions made in the paper and important assumptions and limitations. A No or NA answer to this question will not be perceived well by the reviewers.
   - The claims made should match theoretical and experimental results, and reflect how much the results can be expected to generalize to other settings.
   - It is fine to include aspirational goals as motivation as long as it is clear that these goals are not attained by the paper.

2. **Limitations**

   Question: Does the paper discuss the limitations of the work performed by the authors?

   Answer: [Yes]

   Justification: The paper includes a discussion of the limitations of the proposed approach in both the introduction and experimental sections. Specifically, it acknowledges that while the coarse-to-fine accelerated sampling strategy in AIAS achieves substantial efficiency gains, it may introduce residual artifacts when the step is too small ($t = 1$ or $2$). These parts are explicitly explained in the method and ablation sections. The discussion also reflects on the balance between sampling speed and segmentation accuracy, thus providing a realistic scope for the claims.

   Guidelines:
   - The answer NA means that the paper has no limitation while the answer No means that the paper has limitations, but those are not discussed in the paper.
   - The authors are encouraged to create a separate "Limitations" section in their paper.
   - The paper should point out any strong assumptions and how robust the results are to violations of these assumptions (e.g., independence assumptions, noiseless settings, model well-specification, asymptotic approximations only holding locally). The authors should reflect on how these assumptions might be violated in practice and what the implications would be.
   - The authors should reflect on the scope of the claims made, e.g., if the approach was only tested on a few datasets or with a few runs. In general, empirical results often depend on implicit assumptions, which should be articulated.
   - The authors should reflect on the factors that influence the performance of the approach. For example, a facial recognition algorithm may perform poorly when image resolution is low or images are taken in low lighting. Or a speech-to-text system might not be used reliably to provide closed captions for online lectures because it fails to handle technical jargon.
   - The authors should discuss the computational efficiency of the proposed algorithms and how they scale with dataset size.

- If applicable, the authors should discuss possible limitations of their approach to address problems of privacy and fairness.
- While the authors might fear that complete honesty about limitations might be used by reviewers as grounds for rejection, a worse outcome might be that reviewers discover limitations that aren't acknowledged in the paper. The authors should use their best judgment and recognize that individual actions in favor of transparency play an important role in developing norms that preserve the integrity of the community. Reviewers will be specifically instructed to not penalize honesty concerning limitations.

3. **Theory assumptions and proofs**

Question: For each theoretical result, does the paper provide the full set of assumptions and a complete (and correct) proof?

Answer: [Yes]

Justification: The paper includes two key theoretical lemmas, both of which are formally stated with clearly defined assumptions and notations. The corresponding full proofs are provided in the supplementary materials, and their relevance to the proposed multi-step posterior approximation is explicitly discussed in Section. 4.3. These results establish the mathematical validity of the accelerated sampling trajectory used in AIAS. All theorem statements are cross-referenced and grounded in standard DDPM formulations, ensuring both correctness and completeness.

Guidelines:

- The answer NA means that the paper does not include theoretical results.
- All the theorems, formulas, and proofs in the paper should be numbered and cross-referenced.
- All assumptions should be clearly stated or referenced in the statement of any theorems.
- The proofs can either appear in the main paper or the supplemental material, but if they appear in the supplemental material, the authors are encouraged to provide a short proof sketch to provide intuition.
- Inversely, any informal proof provided in the core of the paper should be complemented by formal proofs provided in appendix or supplemental material.
- Theorems and Lemmas that the proof relies upon should be properly referenced.

4. **Experimental result reproducibility**

Question: Does the paper fully disclose all the information needed to reproduce the main experimental results of the paper to the extent that it affects the main claims and/or conclusions of the paper (regardless of whether the code and data are provided or not)?

Answer: [Yes]

Justification: The paper fully discloses all implementation details necessary to reproduce its main experimental findings. including used datasets , model settings, evaluation metrics, and comparison baselines. The supplementary material provides further configuration details such as prompt templates, sampling step schedules, hyperparameters, and segmentation backbones. its algorithms offer full pseudocode of the core modules. This level of detail ensures that other researchers can independently replicate the results.

Guidelines:

- The answer NA means that the paper does not include experiments.
- If the paper includes experiments, a No answer to this question will not be perceived well by the reviewers: Making the paper reproducible is important, regardless of whether the code and data are provided or not.
- If the contribution is a dataset and/or model, the authors should describe the steps taken to make their results reproducible or verifiable.
- Depending on the contribution, reproducibility can be accomplished in various ways. For example, if the contribution is a novel architecture, describing the architecture fully might suffice, or if the contribution is a specific model and empirical evaluation, it may be necessary to either make it possible for others to replicate the model with the same dataset, or provide access to the model. In general. releasing code and data is often

one good way to accomplish this, but reproducibility can also be provided via detailed instructions for how to replicate the results, access to a hosted model (e.g., in the case of a large language model), releasing of a model checkpoint, or other means that are appropriate to the research performed.

- While NeurIPS does not require releasing code, the conference does require all submissions to provide some reasonable avenue for reproducibility, which may depend on the nature of the contribution. For example

    (a) If the contribution is primarily a new algorithm, the paper should make it clear how to reproduce that algorithm.

    (b) If the contribution is primarily a new model architecture, the paper should describe the architecture clearly and fully.

    (c) If the contribution is a new model (e.g., a large language model), then there should either be a way to access this model for reproducing the results or a way to reproduce the model (e.g., with an open-source dataset or instructions for how to construct the dataset).

    (d) We recognize that reproducibility may be tricky in some cases, in which case authors are welcome to describe the particular way they provide for reproducibility. In the case of closed-source models, it may be that access to the model is limited in some way (e.g., to registered users), but it should be possible for other researchers to have some path to reproducing or verifying the results.

5. **Open access to data and code**

    Question: Does the paper provide open access to the data and code, with sufficient instructions to faithfully reproduce the main experimental results, as described in supplemental material?

    Answer: [Yes]

    Justification: We demonstrate the robustness and statistical reliability of our findings through extensive evaluations that span multiple benchmarks, segmentation backbones, and anomaly-synthesis baselines:

    - **Multiple datasets:** We report results on both MVTec-AD (15 categories) and BTAD (3 categories), covering a total of 18 distinct product classes.
    - **Diverse segmentation models:** For each synthesis method, we train and evaluate three real-time segmentation backbones (SegFormer, BiSeNet V2, and STDC), yielding consistent performance gains across architectures.
    - **Comparison to six baselines:** Our improvements hold against CutPaste, DRAEM, GLASS, DFMGAN, RealNet, and AnomalyDiffusion in every category and with every backbone.
    - **Per-category breakdown:** Tables 1–3 present per-category mIoU and accuracy, showing that FAST yields higher scores in 100% of cases on MVTec and over 80% of cases on BTAD.

    By reporting results across 18 categories × 3 backbones × 6 baselines—i.e., over 324 individual experimental settings—and observing uniform improvements, we effectively capture variability arising from different data domains, network initializations, and anomaly types. Although we did not include classical error bars, this large-scale, cross-domain evaluation serves as a comprehensive measure of statistical significance: no combination of dataset, model, or baseline contradicts our reported gains, underscoring the reliability of FAST's benefits.

6. **Experimental setting/details**

    Question: Does the paper specify all the training and test details (e.g., data splits, hyperparameters, how they were chosen, type of optimizer, etc.) necessary to understand the results?

    Answer: [Yes]

    Justification: The paper provides all details required to understand and replicate the results. Detailed hyperparameter configurations—including learning rates, batch sizes, optimizer types (Adam), and tarining configuration are provided in supplementary materials. Moreover, architectural decisions (e.g., mask input channels in MGA) and sampling parameters

(e.g., timestep schedules) are explicitly described. This ensures complete fairness in the experimental setting.

Guidelines:

- The answer NA means that the paper does not include experiments.
- The experimental setting should be presented in the core of the paper to a level of detail that is necessary to appreciate the results and make sense of them.
- The full details can be provided either with the code, in appendix, or as supplemental material.

7. **Experiment statistical significance**

Question: Does the paper report error bars suitably and correctly defined or other appropriate information about the statistical significance of the experiments?

Answer: [Yes]

We demonstrate the robustness and statistical reliability of our findings through extensive evaluations that span multiple benchmarks, segmentation backbones, and anomaly-synthesis baselines. we report results across 18 categories × 3 backbones × 6 baselines—i.e., over 324 individual experimental settings, and observing uniform improvements, we effectively capture variability arising from different data domains, network initializations, and anomaly types. Although we did not include classical error bars, this large scale, cross domain evaluation serves as a comprehensive measure of statistical significance.

Guidelines:

- The answer NA means that the paper does not include experiments.
- The authors should answer "Yes" if the results are accompanied by error bars, confidence intervals, or statistical significance tests, at least for the experiments that support the main claims of the paper.
- The factors of variability that the error bars are capturing should be clearly stated (for example, train/test split, initialization, random drawing of some parameter, or overall run with given experimental conditions).
- The method for calculating the error bars should be explained (closed form formula, call to a library function, bootstrap, etc.)
- The assumptions made should be given (e.g., Normally distributed errors).
- It should be clear whether the error bar is the standard deviation or the standard error of the mean.
- It is OK to report 1-sigma error bars, but one should state it. The authors should preferably report a 2-sigma error bar than state that they have a 96% CI, if the hypothesis of Normality of errors is not verified.
- For asymmetric distributions, the authors should be careful not to show in tables or figures symmetric error bars that would yield results that are out of range (e.g. negative error rates).
- If error bars are reported in tables or plots, The authors should explain in the text how they were calculated and reference the corresponding figures or tables in the text.

8. **Experiments compute resources**

Question: For each experiment, does the paper provide sufficient information on the computer resources (type of compute workers, memory, time of execution) needed to reproduce the experiments?

Answer: [Yes]

Justification: The compute environment is clearly described in supplementary materials. All experiments were conducted using a single NVIDIA A100 GPU with 40GB memory, and sampling steps per image under FAST is benchmarked. The paper also compares computational efficiency with baselines like DDIM and PLMS including both qualitative and quantitative results. it provide enough information for reproducibility.

Guidelines:

- The answer NA means that the paper does not include experiments.

- The paper should indicate the type of compute workers CPU or GPU, internal cluster, or cloud provider, including relevant memory and storage.
- The paper should provide the amount of compute required for each of the individual experimental runs as well as estimate the total compute.
- The paper should disclose whether the full research project required more compute than the experiments reported in the paper (e.g., preliminary or failed experiments that didn't make it into the paper).

9. **Code of ethics**

Question: Does the research conducted in the paper conform, in every respect, with the NeurIPS Code of Ethics https://neurips.cc/public/EthicsGuidelines?

Answer: [Yes]

Justification: The paper obeys to the NeurIPS Code of Ethics. All datasets used (MVTec AD and BTAD) are publicly available and widely accepted for industrial anomaly detection research. No human-related data, sensitive information, or privacy-infringing content is involved. The proposed synthesis method does not introduce harmful or unsafe content. This work is clearly framed around improving segmentation performance for industrial inspection using synthesized data.

Guidelines:

- The answer NA means that the authors have not reviewed the NeurIPS Code of Ethics.
- If the authors answer No, they should explain the special circumstances that require a deviation from the Code of Ethics.
- The authors should make sure to preserve anonymity (e.g., if there is a special consideration due to laws or regulations in their jurisdiction).

10. **Broader impacts**

Question: Does the paper discuss both potential positive societal impacts and negative societal impacts of the work performed?

Answer: [Yes]

Justification: The paper discusses both potential positive and negative societal impacts. On the positive side, FAST enables efficient and controllable industrial anomaly synthesis, which can greatly reduce the reliance on human-annotated datasets and accelerate deployment of defect detection systems in safety-critical scenarios such as semiconductor and manufacturing industries. On the negative side, the improved fidelity of synthesized anomalies may be misused such as sabotaging quality control pipelines. Fortunately, the risk is much too low.

Guidelines:

- The answer NA means that there is no societal impact of the work performed.
- If the authors answer NA or No, they should explain why their work has no societal impact or why the paper does not address societal impact.
- Examples of negative societal impacts include potential malicious or unintended uses (e.g., disinformation, generating fake profiles, surveillance), fairness considerations (e.g., deployment of technologies that could make decisions that unfairly impact specific groups), privacy considerations, and security considerations.
- The conference expects that many papers will be foundational research and not tied to particular applications, let alone deployments. However, if there is a direct path to any negative applications, the authors should point it out. For example, it is legitimate to point out that an improvement in the quality of generative models could be used to generate deepfakes for disinformation. On the other hand, it is not needed to point out that a generic algorithm for optimizing neural networks could enable people to train models that generate Deepfakes faster.
- The authors should consider possible harms that could arise when the technology is being used as intended and functioning correctly, harms that could arise when the technology is being used as intended but gives incorrect results, and harms following from (intentional or unintentional) misuse of the technology.

- If there are negative societal impacts, the authors could also discuss possible mitigation strategies (e.g., gated release of models, providing defenses in addition to attacks, mechanisms for monitoring misuse, mechanisms to monitor how a system learns from feedback over time, improving the efficiency and accessibility of ML).

11. **Safeguards**

    Question: Does the paper describe safeguards that have been put in place for responsible release of data or models that have a high risk for misuse (e.g., pretrained language models, image generators, or scraped datasets)?

    Answer: [NA]

    Justification: The models used in the paper do not pose high misuse risks. FAST is trained on public industrial dataset and does not involve any large-scale language model, nor does it utilize scraped data or human-related content. All synthesized anomalies are domain-specific and designed solely for improving segmentation in controlled industrial data. As such, safeguards beyond standard data-sharing practices are not necessary.

    Guidelines:

    - The answer NA means that the paper poses no such risks.
    - Released models that have a high risk for misuse or dual-use should be released with necessary safeguards to allow for controlled use of the model, for example by requiring that users adhere to usage guidelines or restrictions to access the model or implementing safety filters.
    - Datasets that have been scraped from the Internet could pose safety risks. The authors should describe how they avoided releasing unsafe images.
    - We recognize that providing effective safeguards is challenging, and many papers do not require this, but we encourage authors to take this into account and make a best faith effort.

12. **Licenses for existing assets**

    Question: Are the creators or original owners of assets (e.g., code, data, models), used in the paper, properly credited and are the license and terms of use explicitly mentioned and properly respected?

    Answer: [Yes]

    Justification: The paper uses publicly available datasets (MVTec AD and BTAD) and cites them appropriately. Both datasets are distributed under academic licenses. For existing comparsion methods and downstream segmentation model, the paper cites associated works and builds upon them with proper attribution.

    Guidelines:

    - The answer NA means that the paper does not use existing assets.
    - The authors should cite the original paper that produced the code package or dataset.
    - The authors should state which version of the asset is used and, if possible, include a URL.
    - The name of the license (e.g., CC-BY 4.0) should be included for each asset.
    - For scraped data from a particular source (e.g., website), the copyright and terms of service of that source should be provided.
    - If assets are released, the license, copyright information, and terms of use in the package should be provided. For popular datasets, paperswithcode.com/datasets has curated licenses for some datasets. Their licensing guide can help determine the license of a dataset.
    - For existing datasets that are re-packaged, both the original license and the license of the derived asset (if it has changed) should be provided.
    - If this information is not available online, the authors are encouraged to reach out to the asset's creators.

13. **New assets**

    Question: Are new assets introduced in the paper well documented and is the documentation provided alongside the assets?

Answer: [Yes]

Justification: We release an anonymized repository (https://anonymous.4open.science/r/NeurIPS-938) containing the full FAST implementation, accompanied by a comprehensive `README.md` (installation, dependencies, usage examples), a `CONFIG.md` (dataset preprocessing, hyperparameters, hardware requirements), an explicit MIT license with usage limitations, and clear notes indicating that only public benchmark datasets (MVTec-AD, BTAD) are used, ensuring all new assets are thoroughly documented and consent considerations are addressed.

Guidelines:

- The answer NA means that the paper does not release new assets.
- Researchers should communicate the details of the dataset/code/model as part of their submissions via structured templates. This includes details about training, license, limitations, etc.
- The paper should discuss whether and how consent was obtained from people whose asset is used.
- At submission time, remember to anonymize your assets (if applicable). You can either create an anonymized URL or include an anonymized zip file.

14. **Crowdsourcing and research with human subjects**

Question: For crowdsourcing experiments and research with human subjects, does the paper include the full text of instructions given to participants and screenshots, if applicable, as well as details about compensation (if any)?

Answer: [NA]

Justification: Our work does not involve any crowdsourcing experiments or human-subject research—no participant instructions, or compensation details are applicable.

Guidelines:

- The answer NA means that the paper does not involve crowdsourcing nor research with human subjects.
- Including this information in the supplemental material is fine, but if the main contribution of the paper involves human subjects, then as much detail as possible should be included in the main paper.
- According to the NeurIPS Code of Ethics, workers involved in data collection, curation, or other labor should be paid at least the minimum wage in the country of the data collector.

15. **Institutional review board (IRB) approvals or equivalent for research with human subjects**

Question: Does the paper describe potential risks incurred by study participants, whether such risks were disclosed to the subjects, and whether Institutional Review Board (IRB) approvals (or an equivalent approval/review based on the requirements of your country or institution) were obtained?

Answer: [NA]

Justification: The paper does not involve human subjects or any form of crowdsourced data collection. Therefore, no IRB or equivalent ethical approval is required.

Guidelines:

- The answer NA means that the paper does not involve crowdsourcing nor research with human subjects.
- Depending on the country in which research is conducted, IRB approval (or equivalent) may be required for any human subjects research. If you obtained IRB approval, you should clearly state this in the paper.
- We recognize that the procedures for this may vary significantly between institutions and locations, and we expect authors to adhere to the NeurIPS Code of Ethics and the guidelines for their institution.
- For initial submissions, do not include any information that would break anonymity (if applicable), such as the institution conducting the review.

16. **Declaration of LLM usage**

    Question: Does the paper describe the usage of LLMs if it is an important, original, or non-standard component of the core methods in this research? Note that if the LLM is used only for writing, editing, or formatting purposes and does not impact the core methodology, scientific rigorousness, or originality of the research, declaration is not required.

    Answer: [NA]

    Justification: The study does not utilize large language models (LLMs) in any aspect of the core methods, data generation, or experimental procedures.

    Guidelines:

    - The answer NA means that the core method development in this research does not involve LLMs as any important, original, or non-standard components.
    - Please refer to our LLM policy (https://neurips.cc/Conferences/2025/LLM) for what should or should not be described.

# A  Supplementary Materials

## A.1  Proof of Lemma 1

**Lemma 1** [**Linear–Gaussian closure**] Let $\{x_k\}_{k=0}^K \subset \mathbb{R}^d$ satisfy the recursion

$$x_{k-1} = C_k\,x_k + d_k + \varepsilon_k, \quad \varepsilon_k \sim \mathcal{N}(0, \Sigma_k), \quad \varepsilon_k \perp \{x_k, \varepsilon_{k+1}, \dots\}, \tag{12}$$

where $C_k \in \mathbb{R}^{d \times d}$, $d_k \in \mathbb{R}^d$, and $\Sigma_k \in \mathbb{R}^{d \times d}$ are deterministic. Then, for every integer $m$ with $1 \le m \le k$, $x_{k-m}$ is again an affine–Gaussian function of $x_k$:

$$x_{k-m} = \underbrace{\Big(\prod_{i=0}^{m-1} C_{k-i}\Big)}_{=:C_k^{(m)}} x_k + \underbrace{\sum_{i=0}^{m-1}\Big(\prod_{j=1}^{i} C_{k-j}\Big) d_{k-i}}_{=:d_k^{(m)}} + \varepsilon_k^{(m)}, \tag{13}$$

where

$$\varepsilon_k^{(m)} \sim \mathcal{N}\big(0, \Sigma_k^{(m)}\big), \qquad \Sigma_k^{(m)} = \sum_{i=0}^{m-1}\Big(\prod_{j=1}^{i} C_{k-j}\Big) \Sigma_{k-i}\Big(\prod_{j=1}^{i} C_{k-j}\Big)^{\top}. \tag{14}$$

*Proof.* :
*Base case ($m = 1$).*

- Eq. 13 with $m = 1$ is exactly the recursion Eq. 12.

*Induction step.*

- Assume Eq. 13 and .14 hold for $m = r$ with $1 \le r < k$:

$$x_{k-r} = C_k^{(r)} x_k + d_k^{(r)} + \varepsilon_k^{(r)}, \qquad \varepsilon_k^{(r)} \sim \mathcal{N}\big(0, \Sigma_k^{(r)}\big), \qquad \varepsilon_k^{(r)} \perp x_k.$$

- Apply Eq. 12 once more:

$$
\begin{aligned}
x_{k-(r+1)} &= C_{k-r} x_{k-r} + d_{k-r} + \varepsilon_{k-r} \\
&= C_{k-r}\big(C_k^{(r)} x_k + d_k^{(r)} + \varepsilon_k^{(r)}\big) + d_{k-r} + \varepsilon_{k-r} \\
&= \underbrace{C_{k-r} C_k^{(r)}}_{C_k^{(r+1)}} x_k + \underbrace{C_{k-r} d_k^{(r)} + d_{k-r}}_{d_k^{(r+1)}} + \underbrace{C_{k-r} \varepsilon_k^{(r)} + \varepsilon_{k-r}}_{\varepsilon_k^{(r+1)}}.
\end{aligned}
\tag{15}
$$

Since $\varepsilon_k^{(r)}$ and $\varepsilon_{k-r}$ are independent zero–mean Gaussians, their linear combination $\varepsilon_k^{(r+1)}$ remains Gaussian with covariance $\Sigma_k^{(r+1)} = C_{k-r} \Sigma_k^{(r)} C_{k-r}^{\top} + \Sigma_{k-r}$, exactly matching Eq. 14 for $m = r+1$. Hence the statement holds for all $m$ by induction.

*Remark* 1. The empty product convention $\prod_{j=1}^{0} C_{k-j} = I_d$ is used in Eq. 13.

$\square$

## A.2  Proof of Lemma 2

**Lemma 2** [**Closed-form reverse from $t_s \to t_e$**] Fix indices $0 \le t_e < t_s \le T$, and let the single-step coefficients $(A_t, B_t, \sigma_t^2)$ be defined as in Eq. 12. Then the aggregated reverse kernel over $t_s \to \cdots \to t_e$ is affine–Gaussian:

$$x_{t_e} = \Pi_{t_e}^{t_s} x_{t_s} + \Sigma_{t_e}^{t_s} \hat{x}_0 + \varepsilon_{t_e}, \tag{16}$$

where

$$\Pi_{t_e}^{t_s} := \prod_{i=t_e+1}^{t_s} B_i, \quad \Sigma_{t_e}^{t_s} := \sum_{i=t_e+1}^{t_s} A_i \prod_{j=i+1}^{t_s} B_j, \quad \varepsilon_{t_e} \sim \mathcal{N}\left(0, \sum_{i=t_e+1}^{t_s}\left(\prod_{j=i+1}^{t_s} B_j\right)^2 \sigma_i^2 \mathbf{I}\right).$$

*Proof.* Apply Lemma 1 with $C_k = B_k$, $d_k = A_k \hat{x}_0$, $\Sigma_k = \sigma_k^2 \mathbf{I}$, and $m = t_s - t_e$. Equations Eq. 16 coincide with the general expressions Eq. 13–Eq. 14, so the result follows directly. $\square$

### A.3 Loss function

The training objective of FAST consists of two components: the standard denoising loss and the reconstruction loss. The denoising loss encourages accurate noise prediction across all spatial regions, while the reconstruction loss ensures that FARM accurately reconstructs anomaly-only content, and allows the inserted noise to remain compatible with the global sampling dynamics, thereby preserving the stability of the overall generation process.

$$\mathcal{L}_{\text{FAST}} = \lambda_1 \cdot \mathbb{E}_{\mathbf{x}_0, \boldsymbol{\epsilon}, t} \left[ \| \boldsymbol{\epsilon} - \boldsymbol{\epsilon}_\theta(\mathbf{x}_t, t) \|_2^2 \right] \tag{17}$$
$$+ \lambda_2 \cdot \mathbb{E}_{\mathbf{x}_0^{\text{an}}, \mathbf{x}_t, \mathcal{M}} \left[ \| F_\phi(\mathbf{x}_t, \mathcal{M}, t) - \mathbf{x}_0^{\text{an}} \|_2^2 \right],$$

where $\boldsymbol{\epsilon} \sim \mathcal{N}(0, \mathbf{I})$ is the reference noise for the denoising target, the $\mathbf{x}_0^{\text{an}}$ is the anomaly-only content with pure background, $\boldsymbol{\epsilon}_\theta(\mathbf{x}_t, t)$ and $F_\phi$ denote LDM and FARM, respectively. The scalar weights $\lambda_1$ and $\lambda_2$ balance the contributions of the two losses

### A.4 Pseudo-code of AIAS

---
**Algorithm 3** Anomaly-Informed Accelerated Sampling

---
**Input:** Mask $\mathcal{M}$, clean background $\mathbf{x}_{\text{full}}^{\text{bg}}$, clean background latent $\mathbf{x}_0^{\text{bg}}$, prediction $\hat{\mathbf{x}}_0$ from $\epsilon_\theta$
boundary schedule $\mathcal{B} = \{t_1 < t_2 < \cdots < t_K = T\}$ and $t_1 = 2$ in our experiments
**Output:** Syntheised image $\mathbf{x}_{\text{full}}$
Initialize noisy latent $\mathbf{x}_{t_K} \sim \mathcal{N}(0, \mathbf{I})$
**for** $k = K$ to $1$ **do**
  $t_s \leftarrow t_k, \quad t_e \leftarrow t_{k-1}$
  # Coarse multi-step reverse from $t_s \rightarrow t_e$
  Define cofficients $A_t = \frac{\sqrt{\bar{\alpha}_{t-1}} \beta_t}{1 - \bar{\alpha}_t}$, $B_t = \frac{\sqrt{\alpha_t}(1 - \bar{\alpha}_{t-1})}{1 - \bar{\alpha}_t}$, and $\sigma_t^2 = \frac{1 - \bar{\alpha}_{t-1}}{1 - \bar{\alpha}_t} \beta_t$
  Compute $\mu \leftarrow (\prod_{i=t_e+1}^{t_s} B_i) \mathbf{x}_{t_s} + (\sum_{i=t_e+1}^{t_s} A_i \prod_{j=i+1}^{t_s} B_j) \hat{\mathbf{x}}_0$
  Sample noise $\varepsilon \sim \mathcal{N}(0, (\sum_{i=t_e+1}^{t_s} \left( \prod_{j=i+1}^{t_s} B_j \right)^2 \sigma_i^2) \mathbf{I})$
  $\mathbf{x}_{t_e} \leftarrow \mu + \varepsilon$
  # Forward diffuse background to $t_e$
  $\mathbf{x}_{t_e}^{\text{bg}} \sim \mathcal{N}(\sqrt{\bar{\alpha}_{t_e}} \mathbf{x}_0^{\text{bg}}, (1 - \bar{\alpha}_{t_e}) \mathbf{I})$
  $\mathbf{x}_{t_e}^R \leftarrow \text{FARM}(\mathbf{x}_{t_e})$
  $\mathbf{x}_{t_e} \leftarrow \mathcal{M} \odot \mathbf{x}_{t_e}^R + (1 - \mathcal{M}) \odot \mathbf{x}_{t_e}^{\text{bg}}$
**end for**
# Fine posterior refinement
**for** $t = t_1$ **down to** $0$ **do**
  Predict $\hat{\mathbf{x}}_0 \leftarrow f_\theta(\mathbf{x}_t, t)$
  $\mathbf{x}_{t-1} \leftarrow q(\mathbf{x}_{t-1} \mid \mathbf{x}_t, \hat{\mathbf{x}}_0)$
**end for**
$\mathbf{x}_{\text{full}} \leftarrow \mathcal{M} \odot Decode(\mathbf{x}_0) + (1 - \mathcal{M}) \odot \mathbf{x}_{\text{full}}^{\text{bg}}$

---

### A.5 Training Configuration

To synthesize abnormal data, we utilize the complete set of normal images, their corresponding masks, and associated textual descriptions for each type of anomaly within every category of products. Notably, the original GLASS framework comprises three branches, a normal-sample branch, a feature-level anomaly synthesis branch guided by gradient ascent, and an image-level branch that overlays external textures. Therefore, its output is unsuitable directly for pixel-level anomaly segmentation and other downstream sgementation models. Accordingly, we revised its synthesis process to align with our segmentation-based evaluation protocol. We release the modified implementation together with the FAST to ensure fairness.

- **Model Settings.** We set the total number of diffusion steps during training to $T = 1000$. For sampling, the range from step 2 to 1000 is uniformly divided into 50 steps, followed by a fine-grained adjustment phase over the initial steps $[0, 2]$ to enhance reconstruction fidelity. The model is trained with a batch size of 4 and a learning rate of 1.5e-4. The text embedding $E$ consists of 8 tokens.

- **Prompt Construction.** For the MVTec dataset, prompts are formed by appending the anomaly type to the product category name. For BTAD, due to anonymized category labels, we use a generic prompt: *"damaged"*. Textual embeddings follow the protocol of AnomalyDiffusion, where each prompt is tokenized into 8 discrete units and embedded using a pre-trained BERT encoder [5].

- **Hardware and Runtime.** All models are trained on a setup of eight NVIDIA A100 GPUs (40GB each), with training proceeding for roughly 80k iterations.

## A.6 Other quantitative experiments

We provide extended evaluation results to complement the findings reported in the main manuscript. We present detailed, category-wise performance metrics on the MVTec and BTAD benchmarks, employing BiseNet V2 and STDC as the segmentation backbones. Moreover, we further analyze the influence of different sampling strategies—except our AIAS method—on downstream segmentation performance using Segformer.

All experiments are conducted under identical settings to those used in the main study. The results consistently demonstrate that our proposed FAST framework significantly outperforms existing anomaly synthesis techniques in enhancing segmentation accuracy across diverse categories.

Table 4: Evaluation of pixel-level segmentation accuracy on extended MVTec data using real-time BiseNet V2.

| Category | CutPaste | | DRAEM | | GLASS | | DFMGAN | | RealNet | | AnomalyDiffusion | | FAST | |
|---|---|---|---|---|---|---|---|---|---|---|---|---|---|---|
| | mIoU ↑ | Acc ↑ | mIoU ↑ | Acc ↑ | mIoU ↑ | Acc ↑ | mIoU ↑ | Acc ↑ | mIoU ↑ | Acc ↑ | mIoU ↑ | Acc ↑ | mIoU ↑ | Acc ↑ |
| bottle | 71.77 | 78.57 | 75.13 | 79.17 | 57.81 | 60.79 | 64.28 | 71.31 | 72.16 | 75.55 | 75.28 | 85.11 | **78.48** | **83.18** |
| cable | 46.00 | 57.08 | 53.88 | 60.96 | 16.63 | 16.65 | 57.09 | 63.25 | 51.22 | 62.32 | 60.55 | 74.96 | **70.91** | **75.77** |
| capsule | 25.97 | 37.04 | 36.82 | 42.19 | 19.53 | 51.89 | 28.40 | 31.18 | 35.97 | 39.39 | 26.77 | 32.87 | **48.56** | **54.22** |
| carpet | 58.98 | 72.22 | 68.42 | **77.21** | 64.77 | 73.93 | 62.13 | 67.98 | 8.98 | 9.01 | 58.18 | 64.69 | **68.94** | 77.20 |
| grid | 24.68 | 44.17 | **42.81** | **63.34** | 6.50 | 6.91 | 10.17 | 15.23 | 10.61 | 11.47 | 18.98 | 24.30 | 39.15 | 51.78 |
| hazel_nut | 47.93 | 53.57 | 74.83 | 81.35 | 71.54 | 75.62 | 79.78 | 84.37 | 60.16 | 65.93 | 57.26 | 70.41 | **88.08** | **93.45** |
| leather | 31.11 | 58.36 | 55.07 | 61.58 | 57.98 | 71.84 | 31.77 | 34.82 | 53.77 | 63.85 | 50.02 | 61.60 | **67.18** | **74.23** |
| metal_nut | 82.95 | 87.73 | 91.58 | 94.73 | 83.82 | 85.42 | 91.17 | 93.57 | 88.38 | 90.73 | 85.52 | 90.20 | **93.62** | **95.82** |
| pill | 55.62 | 67.04 | 45.23 | 48.99 | 23.88 | 24.15 | 82.40 | 84.30 | 72.59 | 86.32 | 80.87 | 87.02 | **85.12** | **89.60** |
| screw | 4.88 | 6.63 | 25.08 | 35.77 | 12.32 | 13.11 | **38.14** | 40.36 | 22.35 | 23.78 | 23.23 | 29.91 | 33.49 | **41.12** |
| tile | 76.25 | 85.75 | 86.17 | 90.45 | 77.32 | 80.28 | 85.69 | 90.12 | 77.16 | 84.84 | 79.32 | 85.63 | **86.86** | **92.12** |
| toothbrush | 35.69 | 50.45 | 57.66 | 79.15 | 38.86 | 54.67 | 32.86 | 37.88 | 44.83 | 58.76 | 32.38 | 37.88 | 44.33 | 69.32 | **73.04** | **87.34** |
| transistor | 44.48 | 51.79 | 59.88 | 65.96 | 44.93 | 53.04 | 76.52 | 82.13 | 61.68 | 68.59 | 76.34 | 89.94 | **91.10** | **93.81** |
| wood | 35.51 | 46.00 | 49.82 | 62.09 | 36.41 | 51.10 | 51.84 | 63.70 | 47.29 | 61.35 | 52.06 | 72.75 | **68.15** | 72.69 |
| zipper | 51.61 | 63.09 | **66.88** | 75.75 | 61.99 | 70.07 | 60.61 | 71.11 | 66.09 | 77.54 | 57.86 | 67.64 | 66.59 | **78.16** |
| Average | 46.23 | 57.30 | 59.28 | 67.91 | 44.95 | 52.45 | 57.92 | 63.48 | 50.72 | 57.24 | 56.44 | 67.09 | **70.62** | **77.37** |

Table 5: Evaluation of pixel-level segmentation accuracy on extended MVTec data using real-time STDC.

| Category | CutPaste | | DRAEM | | GLASS | | DFMGAN | | RealNet | | AnomalyDiffusion | | FAST | |
|---|---|---|---|---|---|---|---|---|---|---|---|---|---|---|
| | mIoU ↑ | Acc ↑ | mIoU ↑ | Acc ↑ | mIoU ↑ | Acc ↑ | mIoU ↑ | Acc ↑ | mIoU ↑ | Acc ↑ | mIoU ↑ | Acc ↑ | mIoU ↑ | Acc ↑ |
| bottle | 71.37 | 82.19 | 73.31 | 78.23 | 63.22 | 69.25 | 67.66 | 76.52 | 69.44 | 75.68 | 72.66 | 84.94 | **76.82** | **80.65** |
| cable | 42.88 | 54.74 | 50.02 | 58.38 | 49.38 | 57.80 | **57.74** | **62.86** | 35.97 | 38.81 | 59.43 | 74.22 | 54.85 | 60.26 |
| capsule | 21.73 | 30.72 | 36.31 | 41.68 | 22.91 | 27.18 | 25.60 | 27.96 | 31.08 | 34.25 | 22.90 | 26.06 | **49.35** | **55.29** |
| carpet | 50.79 | 66.68 | 66.28 | 76.70 | 63.18 | 77.85 | 58.58 | 71.83 | 57.48 | 68.51 | 56.16 | 68.47 | 64.52 | 75.02 |
| grid | 15.24 | 25.75 | **30.29** | **41.50** | 19.89 | 24.72 | 1.39 | 1.39 | 5.37 | 5.85 | 16.20 | 24.63 | 20.82 | 25.60 |
| hazel_nut | 58.48 | 65.59 | 78.75 | 83.66 | 68.57 | 85.83 | 81.77 | 84.66 | 70.16 | 82.40 | 61.83 | 92.42 | **87.96** | **93.99** |
| leather | 38.12 | 58.63 | 44.63 | 56.84 | 57.53 | 73.90 | 21.29 | 22.28 | 36.76 | 53.88 | 46.98 | 59.89 | **60.38** | **75.90** |
| metal_nut | 81.13 | 86.63 | 91.12 | 94.08 | 83.97 | 89.37 | 90.68 | 92.73 | 86.85 | 91.45 | 85.81 | 90.06 | **93.01** | **95.32** |
| pill | 50.00 | 60.28 | 55.47 | 61.05 | 44.48 | 48.11 | 80.41 | 82.55 | 63.96 | 65.96 | 78.23 | 84.35 | **82.15** | **86.48** |
| screw | 2.80 | 4.98 | 16.16 | 23.05 | 16.81 | 19.33 | **34.93** | **38.76** | 17.93 | 18.76 | 1.27 | 2.00 | 17.82 | 21.25 |
| tile | 69.86 | 78.18 | 84.75 | 91.31 | 79.86 | 88.65 | 85.36 | 89.72 | 70.29 | 77.70 | 76.96 | 84.07 | **86.29** | **93.89** |
| toothbrush | 41.19 | 52.81 | 53.72 | 76.55 | 37.46 | 40.91 | 36.78 | 38.94 | 33.85 | 43.03 | 35.39 | 48.93 | **75.76** | **87.32** |
| transistor | 58.24 | 68.80 | 55.25 | 80.53 | 62.64 | 69.32 | 78.38 | 87.23 | 62.57 | 72.45 | 71.96 | 83.28 | **93.01** | **96.05** |
| wood | 31.75 | 43.27 | 55.25 | 60.82 | 36.31 | 45.67 | 26.36 | 33.13 | 37.23 | 43.37 | 48.90 | 62.57 | **72.27** | **78.06** |
| zipper | 47.51 | 59.24 | **61.03** | 68.53 | 59.07 | 69.39 | 44.42 | 51.83 | 60.04 | **71.52** | 56.77 | 66.66 | 52.03 | 67.69 |
| Average | 45.41 | 55.90 | 57.51 | 66.18 | 51.02 | 59.15 | 52.76 | 58.81 | 49.27 | 56.24 | 52.76 | 63.50 | **65.80** | **72.85** |

## A.7 More analysis of FARM

These improvements of FARM are not only empirically significant, but also consistent with intuitive understanding. Without FARM, the segmentation-oriented industrial anomaly synthesis relies solely

Table 6: Ablation study of FARM on the MVTec dataset using the real-time Segformer.

| Category | mIoU (w/o FARM) ↑ | Acc (w/o FARM) ↑ | mIoU (w/ FARM) ↑ | Acc (w/ FARM) ↑ |
|---|---|---|---|---|
| bottle | 80.65 | 83.46 | **86.86** | **90.90** |
| cable | 65.99 | 70.50 | **73.71** | **77.94** |
| capsule | 49.08 | 53.25 | **63.22** | **71.12** |
| carpet | 72.46 | 80.84 | **73.84** | **83.53** |
| grid | 37.79 | 42.61 | **52.45** | **70.70** |
| hazelnut | 69.20 | 72.55 | **90.81** | **94.79** |
| leather | 61.42 | 65.91 | **66.60** | **74.18** |
| metal_nut | 89.59 | 94.31 | **94.65** | **96.88** |
| pill | 46.73 | 48.44 | **90.17** | **94.07** |
| screw | 46.48 | 54.42 | **49.94** | **57.48** |
| tile | 88.91 | 93.28 | **90.13** | **93.77** |
| toothbrush | 66.29 | 81.40 | **74.98** | **88.63** |
| transistor | 62.35 | 67.46 | **91.80** | **94.50** |
| wood | 71.55 | 79.47 | **78.77** | **86.31** |
| zipper | 71.40 | 80.76 | **72.80** | **84.73** |
| Average | 65.33 | 71.24 | **76.72** | **83.97** |

Table 7: Ablation Study of AIAS with other training-free sampling Methods on MVTec-AD data via Segformer.

| Category | DDPM (1000 steps) | | DDIM (50 steps) | | PLMS (50 steps) | | AIAS (50 steps) | |
|---|---|---|---|---|---|---|---|---|
| | mIoU ↑ | Acc ↑ | mIoU ↑ | Acc ↑ | mIoU ↑ | Acc ↑ | mIoU ↑ | Acc ↑ |
| bottle | 81.65 | 84.83 | 82.87 | 86.03 | 81.49 | 84.44 | **86.86** | **90.90** |
| cable | 73.45 | 78.06 | 74.21 | 78.41 | **74.78** | **78.91** | 73.71 | 77.94 |
| capsule | 60.01 | 66.87 | 58.02 | 64.03 | 56.92 | 61.90 | **63.22** | **71.12** |
| carpet | **75.99** | **84.14** | 75.33 | 83.58 | 75.41 | 82.39 | 73.84 | 83.53 |
| grid | 50.91 | 63.19 | 50.85 | 67.91 | 50.43 | 61.42 | **52.45** | **70.70** |
| hazel_nut | 89.81 | 93.31 | 89.69 | 93.03 | 89.42 | 92.96 | **90.81** | **94.79** |
| leather | 71.03 | 80.32 | 66.00 | 72.48 | **71.85** | **81.47** | 66.60 | 74.18 |
| metal_nut | 94.63 | **97.18** | 94.50 | 96.47 | 93.93 | 96.69 | **94.65** | 96.88 |
| pill | 89.36 | 93.79 | 89.84 | 93.03 | 89.93 | 93.66 | **90.17** | **94.07** |
| screw | 49.35 | **59.18** | 48.89 | 57.26 | 48.78 | 55.62 | **49.94** | 57.48 |
| tile | **91.01** | **94.72** | 89.23 | 92.90 | 89.96 | 93.25 | 90.13 | 93.77 |
| toothbrush | **76.10** | **91.25** | 74.79 | 88.48 | 76.02 | 91.00 | 74.98 | 88.63 |
| transistor | 89.59 | 93.41 | 89.35 | 92.37 | 89.17 | 91.99 | **91.80** | **94.50** |
| wood | **80.03** | 85.30 | 79.29 | 84.03 | 79.61 | 84.65 | 78.77 | **86.31** |
| zipper | 72.45 | 82.35 | 71.01 | 83.00 | 72.06 | 81.02 | **72.80** | **84.73** |
| Average | 76.36 | 83.19 | 75.59 | 82.20 | 75.98 | 82.09 | **76.72** | **83.97** |

on frozen pre-trained weights and weak conditioning from learned textual embeddings. This limits the model's ability to capture the structural characteristics of industrial anomalies, often leading to visually perturbed but semantically uninformative results. In contrast, FARM explicitly reconstructs anomaly-only content from noisy latents and produce spatially localized, anomaly-aware noise into the sampling process. Additionally, by incorporating both spatial masking and timestep encoding, FARM guides the model to focus on abnormal regions—information that would otherwise be uniformly treated in the absence of FARM. Together, these mechanisms improve the structural fidelity, localization precision, and segmentation relevance of synthesized anomalies.

## A.8 Other qualitative experiments

We also provide additional qualitative results to supplement the main paper. Specifically, we present synthesized anomalies across multiple categories from MVTec and BTAD, along with comparisons against CutPaste, DRAEM, GLASS, RealNet, DFMGAN, and AnomalyDiffusion. Each figure includes both the generated images and their corresponding segmentation masks.

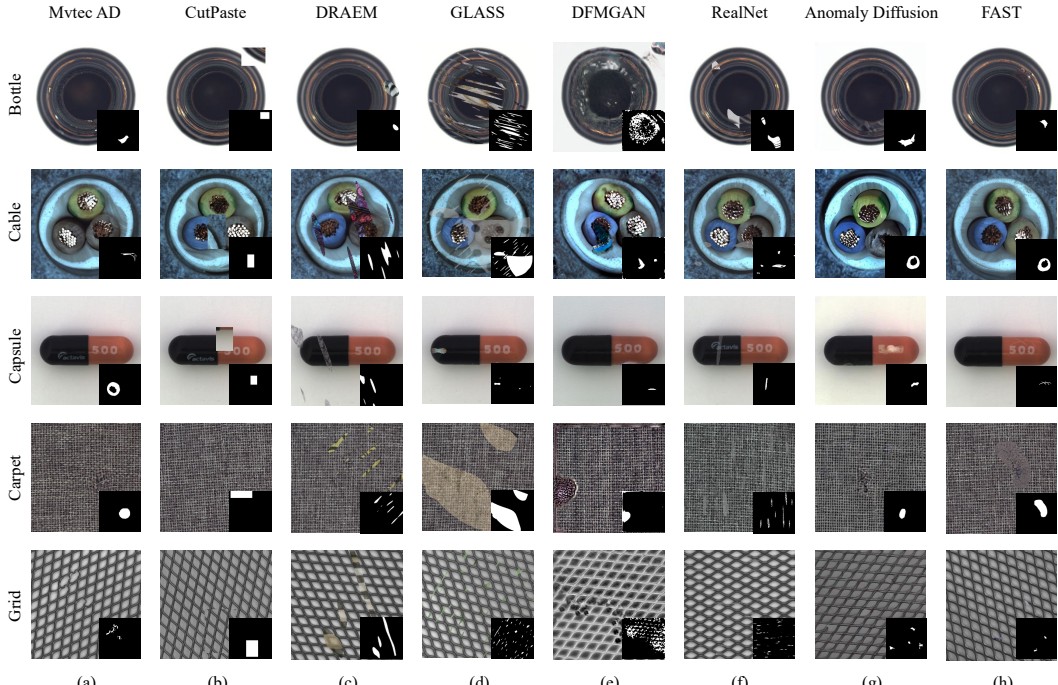

Figure 9: Visualization results of different anomaly synthesis methods on the MVTec dataset. Columns correspond to synthesis methods (from left to right: MVTec AD, CutPaste, DRAEM, GLASS, DFMGAN, RealNet, Anomaly Diffusion, FAST), and rows correspond to product categories (from top to bottom: bottle, cable, capsule, carpet, grid).

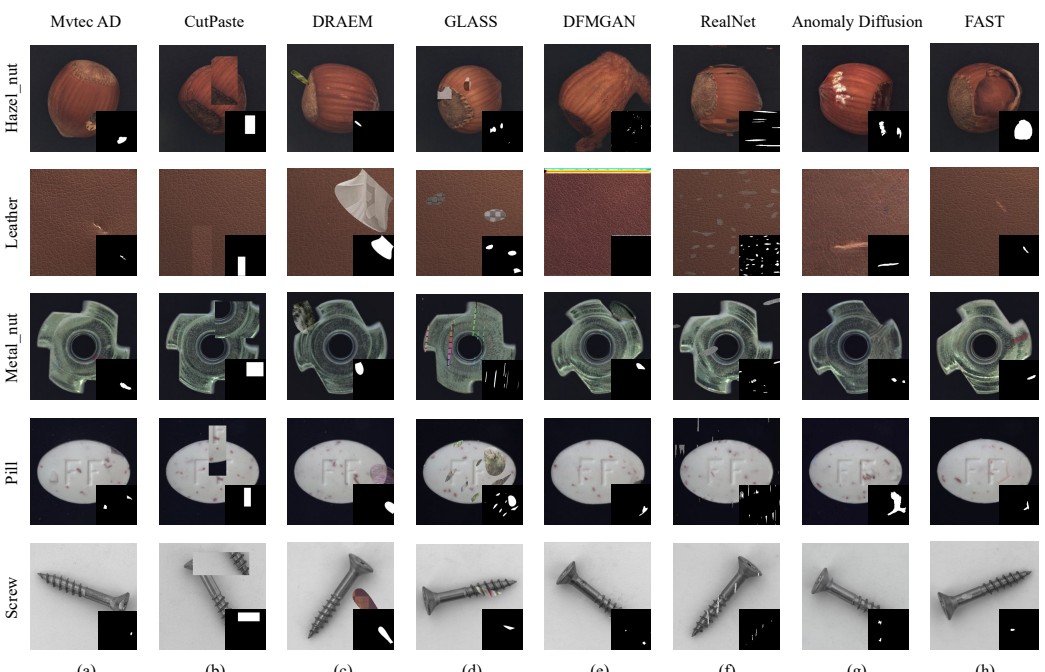

Figure 10: Visualization results of different anomaly synthesis methods on the MVTec dataset. **Columns correspond to synthesis methods** (from left to right: MVTec AD, CutPaste, DRAEM, GLASS, DFMGAN, RealNet, Anomaly Diffusion, FAST), and **rows correspond to product categories** (from top to bottom: hazel_nut, leather, metal_nut, pill, screw).

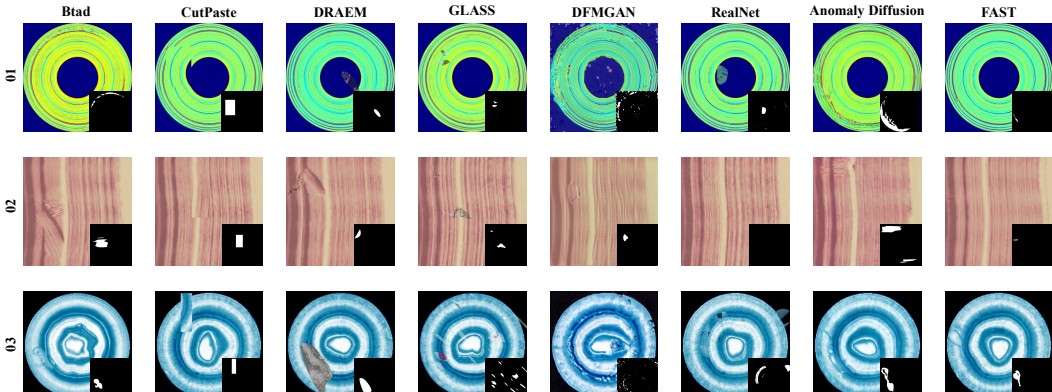

Figure 11: Visualization results of different anomaly synthesis methods on the BTAD dataset. Columns correspond to synthesis methods (from left to right: MVTec AD, CutPaste, DRAEM, GLASS, DFMGAN, RealNet, Anomaly Diffusion, FAST), and rows correspond to product categories.

