# OpenReview forum: "FAST: Foreground‑aware Diffusion with Accelerated Sampling Trajectory for Segmentation‑oriented Anomaly Synthesis"
_NeurIPS.cc/2025/Conference — NeurIPS 2025 poster_

### Official Review · Reviewer_Wznz · 2025-06-30

**Clarity:** 3
**Significance:** 3
**Originality:** 3
**Rating:** 4
**Confidence:** 4

**Summary:**

This paper proposes FAST, a diffusion-based framework for generating industrial anomaly synthesis data for segmentation tasks. The framework consists of two main components: first, AIAS, a training-free sampling strategy that reduces the denoising steps from the DDPM standard of 1000 to 10, and second, FARM, a module that reconstructs anomaly-specific content from noisy inputs and injects anomaly-aware noise at each sampling step to preserve anomaly signals throughout the overall denoising process. The authors evaluate their method by generating 500 synthetic samples per category on two datasets and training multiple segmentation models with the synthetic data, achieving performance improvements.

**Questions:**

read weakness please

**Ethical Concerns:**

["NO or VERY MINOR ethics concerns only"]

**Final Justification:**

I appreciate the authors' clear rebuttal. While you have addressed some of my concerns well, some concerns have only been partially resolved, so I have maintained my score in order to maintain a positive evaluation of the paper.

**Limitations:**

The core approximation of AIAS relies on assumptions that may not hold in practice, particularly with complex noise schedules or highly variable x^_0 predictions. The theoretical analysis of approximation error bounds and convergence properties is limited, and the error accumulation patterns according to segment length are not sufficiently characterized. And it is questionable whether reducing sampling steps to 10 is truly a priority in synthetic data generation. In industrial deployment, final segmentation performance may be more important than generation speed, and the potential quality degradation at extremely few steps has not been sufficiently analyzed.

**Paper Formatting Concerns:**

1. In figure 1 caption :  foregrond-aware needs to change for foreground-aware
2. Main Paper and Supplementary Material should maintain independent equation numbering systems (Algorithm 3, Equation 13 in Main paper).
3.Additionally, abbreviated forms such as "Sec. 4.3" and full forms such as "Section 4.3" are used inconsistently throughout the paper.

**Quality:**

3

**Strengths And Weaknesses:**

Strength
1. AIAS provide well-formulated theorems that justify the multi-step backward approximation. The supplementary material clearly explains the proof and derivation process for this procedure.
2. There is computational efficiency by reducing sampling steps to around 10 during the generation phase.
3. Systematic comparison and justification through various ablation studies.
4. FARM's foreground-aware processing tends to address the uniform spatial processing limitation of existing diffusion models.

Weaknesses
1. It is questionable whether the characteristic feature of this paper - reducing sampling steps to around 10 - is truly a significant advantage for synthetic data generation. In industrial anomaly detection, fast inference is important only during the anomaly segmentation stage, but for synthetic data generation, wouldn't quality be more important? Wouldn't it be better to keep only the foreground-aware processing and use multiple steps for sampling?
2. While the fixed x̂_0 assumption is somewhat acknowledged, isn't it the case that x̂_0 actually changes quite significantly over 10 steps? This raises concerns since Lemma 2 relies on this assumption.
3. In the experimental phase, it was mentioned that 500 synthetic samples were generated, with 1/3 used for training and the rest for evaluation, but a clearer explanation seems necessary regarding the methods used, including masks.
4. In core multi-step aggregation formula x_te = Πx_ts + Σx̂_o + ε_te of AIAS, there is insufficient theoretical analysis of how approximation errors accumulate with respect to segment length (t_s - t_e). Information about how errors are amplified in the actual diffusion process seem necessary.

---

> ### Author Rebuttal · Authors · 2025-07-30
>
> We sincerely thank you for your detailed comments and positive feedback, such as *"clear theoretical justification" and "complete studies."* Below are our responses to your questions.
>
> **Q1: Practical benefit of fast sampling in anomaly synthesis.**
>
> We thank you for this insightful comment. While synthesis quality is important, we argue that fast sampling is highly beneficial in segmentation-oriented industrial anomaly synthesis (SIAS).
>
> 1. In real-world manufacturing, anomaly types evolve quickly, and segmentation models must be updated frequently. FAST enables engineers to generate hundreds of segmentation-aligned anomalies per class in minutes rather than hours or days after training, making rapid retraining feasible.
> 2. Our proposed AIAS achieves up to 100× speedup by reducing sampling steps from 1000 to 10, while still maintaining competitive segmentation performance (see Table 3 in the paper).
> 3. We observe that longer sampling may enhance visual realism but introduce overfitting to generative details at the cost of anomaly segmentation consistency. Therefore, in our context, slightly trading off fine-grained fidelity for significant speed gains is not only acceptable but necessary for practical deployment.
>
> ---
>
> **Q2: Detailed experimental setup.**
>
> We appreciate your suggestion and provide further explanation.
>
> In the data synthesis phase, we follow AnomalyDiffusion’s mask generation strategy by employing two complementary methods:
> (1) augmenting real anomaly masks via geometric and morphological transformations, and
> (2) training a Latent Diffusion Model to learn the distribution of real masks and generate novel ones.
> All generated masks are screened to ensure plausibility before being used to synthesize anomaly–mask pairs. Detailed information can be found in the AnomalyDiffusion paper. For each anomaly type within every object class, we generate 500 anomaly–mask pairs to ensure sufficient diversity for segmentation training.
>
> In the downstream segmentation task, we employ SegFormer, BiseNetV2 and STDC as backbone models, as they are widely adopted in industrial scenarios due to their balance of accuracy and efficiency. Besides, we uniformly split the real anomalies from MVTec and BTAD into 2/3 for validation and 1/3 for training. The training set of segmentation models combines this 1/3 subset with the synthetic samples. This design ensures that the segmentation model is evaluated solely on unseen data while benefiting from synthetic samples during training. Additional implementation details, including model settings and training parameters, are available in Supplementary Materials A.5.
>
> ---
>
> **Q3: The validity of the fixed $\hat{x}_0$ assumption in Lemma 2.**
>
> We appreciate your concern and provide the following reasons for the fixed $\hat{x}_0$ assumption.
>
> 1. **Training Objective Consistency:**
> The denoising diffusion model's training objective directly encourages consistency in predicting $\hat{x}_0$ across different timesteps, particularly at lower $t$. This ensures that even when reused in a large segment, a fixed $\hat{x}_0$ is still relatively accurate.
>
> 2. **AIAS Theoretical support:**
> In AIAS, the reverse process is divided into limited segments. Within each segment, the trajectory is approximated by a fixed  $\hat{x}_0$ to maintaini the denoising distribution.  At the start of each segment, the network re-calibrates  $\hat{x}_0$ based on $x_t$  from the previous segment, effectively preventing error accumulation across segments. Previous work such as diffusion-based anomaly detection methods, also demonstrates that diffusion models are robust to small perturbations in $\hat{x}_0$, especially when guided by strong cues like masks or backgrounds. Thus, even if $\hat{x}_0$ from the previous segment is slightly biased, the trajectory remains aligned with the data manifold.
>
> 3. **Denoising Distribution Dependence:**
> The denoising distribution is Gaussian, with its mean given by the posterior mean:
>
> &emsp;&emsp; $\mu_{t-1 \mid t} = A_t x_0 + B_t x_t$
>
> &emsp;&emsp; where the coefficient $A_t$ is defined as
>
> &emsp;&emsp; $A_t = \frac{\sqrt{\bar{\alpha}_{t-1}} \, \beta_t}{1 - \bar{\alpha}_t}$.
>
> &emsp;&emsp; and the coefficient $B_t$ is defined as
>
> &emsp;&emsp; $B_t = \frac{\sqrt{\alpha_t}(1 - \bar{\alpha}_{t-1})}{1 - \bar{\alpha}_t}$.
>
> &emsp;&emsp; As $t$ decreases, $\hat{x}_0$ becomes more reliable, with $A_t$ increasing and $B_t$ decreasing. Therefore, AIAS increasingly relies on the re-calibrated $\hat{x}_0$, which helps mitigate the propagation of error from $x_t$. This also motivates the design of AIAS as a coarse-to-fine aggregation process.
>
> 4. **Experimental Results:**
> Our experiments show that increasing $K$ actually provides noticeable improvements within a certain range. However, with $K=10$, the results are already acceptable.
>
> ---
> **Q4: Theoretical analysis of error accumulation.**
>
> Thank you for your suggestion. The sampling approximation error in AIAS originates from two sources:
> (1) *local segment-wise error*, introduced by keeping $\hat{x}_0$ fixed within each segment, and
> (2) *global cumulative error*, resulting from propagation across segments.
>
> Ideally $\hat{x}_0^{(t)}$ should be adaptively estimated **at each time step t**.
>
> However, AIAS holds $\hat{x}^k_0$ fixed within the  $k_{th}$ segment, leading to a local approximation error at the endpoint $t_e$.
> $$
> \delta_{t_e} = \Delta_{t \rightarrow t_e} := \Sigma (\hat{x}_0 - \hat{x}_0^{(t)})
> \tag{1}
> $$
>
> Assuming the full reverse process is divided into $K$ segments with partition points:
> $$
> T = t_0 > t_1 > t_2 > \cdots > t_K = 0,
> $$
> Each segment $(t_k \rightarrow t_{k+1})$ contributes a local error and the global cumulative error can be described recursively. Specifically, the offset at time $t_1$ is computed as:
> $\delta_{t_1} = \Sigma_0 (\hat{x}_0^{(0)} - \hat{x}_0^{(t_0)})$.
>
> Then, the state at $t_2$ follows:
> $x_{t_2} = \Pi_1 x_{t_1} + \Sigma_1 \hat{x}_0^{(1)}$,
>
> which leads to the offset propagation:
> $\delta_{t_2} = \Pi_1 \delta_{t_1} + \Sigma_1 (\hat{x}_0^{(1)} - \hat{x}_0^{(t_1)})$.
>
> By recursion, the total error at the final timestep $t_K$ can be expressed as:
> $$
> \Delta_{\text{total}} = \delta_{t_K} = \sum_{k=0}^{K-1} \left( \prod_{l = k+1}^{K-1} \Pi_l \right) \cdot \Sigma_k \left( \hat{x}_0^{(k)} - \hat{x}_0^{(t_k)} \right).
> \tag{2}
> $$
>
> We further assume that the prediction error of $\hat{x}_0$ is uniformly bounded at each segment:
> - $\|\hat{x}_0^{(k)} - \hat{x}_0^{(t_k)}\| \leq \epsilon_0$
> - $\|\Sigma_k\| \leq C_\Sigma$
> - $\|\Pi_k\| \leq \alpha < 1$
>
> Here, $\alpha$ represents the maximum norm of the inter-segment state propagation matrices $\Pi_k$, and $C_\Sigma$ denotes the influence of $\hat{x}_0$ on the subsequent state within a segment. Notably, **both are deterministic and computable under known noise schedules, as shown in Lemma 1**.
>
> Therefore, the total error admits the following conservative upper bound:
> $$
> \|\Delta_{\text{total}}\| \leq \sum_{k=0}^{K-1} \alpha^{K-1-k} \cdot C_\Sigma \epsilon_0
> = C_\Sigma \epsilon_0 \sum_{m=0}^{K-1} \alpha^m
> = C_\Sigma \epsilon_0 \cdot \frac{1 - \alpha^K}{1 - \alpha}.
> \tag{3}
> $$
>
> **This bound is highly conservative in practice**, $\epsilon_0$ tends to decrease as $K$ increases, since shorter segments reduce approximation error and the overall procedure becomes closer to standard DDPM sampling. Moreover, the assumption that all errors accumulate in the same direction is pessimistic. Due to the inherent robustness of denoising diffusion models, the error introduced in one segment is typically corrected in the next, as $\hat{x}_0$ is re-estimated from the updated $x_t$. This re-calibration helps mitigate the propagation of prior inaccuracies.
>
>
> These observations further explain why AIAS remains effective even when using only $K = 10$ segments. The combined effect of theoretical error control, diffusion model robustness, and segment-wise re-estimation of $\hat{x}_0$ ensures that the synthetic samples retain acceptable fidelity.

---

> > ### Comment · Reviewer_Wznz · 2025-08-06
> > **Comment and Concerns**
> >
> > Q1 : While I understand the benefits of rapid data generation post-training, I still believe inference speed is more critical. Even though your method is model-agnostic, shouldn't quality take precedence over generation speed if segmentation performance is the ultimate goal? Frankly, I'm not convinced that segmentation models are updated so frequently in real manufacturing settings, and even if they are updated, I cannot imagine scenarios where existing model inference would be shut down.
> > Q2-Q4:I believe the concerns have been addressed to some extent.
> > Additionally, after reading other reviewers' comments, I am curious about results on standard evaluation metrics for industrial anomaly detection papers such as AUROC and PRO. Therefore, while I acknowledge the technical contributions, I have questions about practical priorities and the lack of standard evaluation metrics.

---

> ### Author Response · Authors · 2025-08-07
>
> We thank the reviewer for the follow-up comment. We provide response below for the raised questions.
> ### Sampling speed and anomaly quality.
> We acknowledge the importance of generation quality, but we want reiterate that FAST is specifically designed for **Segmentation-Oriented Industrial Anomaly Synthesis (SIAS)**, which differs from traditional anomaly detection or generic image generation.
> In SIAS, the synthesized data is used to adapt segmentation models. Our goal is to generate structurally aligned, mask-consistent anomalies in a timely manner, rather than focusing on visual realism. That is why using too large steps leads to degraded segmentation performance. While increasing the number of steps within a certain range can marginally improve image quality, it comes at the cost of significantly longer inference time, as shown in our ablation study.
> Besides, in real-world manufacturing, it is common for production lines to undergo frequent reconfiguration (known as production line changeover) to accommodate new products or materials. These changes often introduce new types of anomalies that existing segmentation models cannot handle. Delays in updating segmentation models during such transitions can lead to increased manual inspections, false positives, missed detections, and even line downtime. FAST is particularly suitable for solving this issue. As far as we know, there are also domain adaptation studies that focus on this issue, aiming to reduce the time required for segmentation model adaptation in order to improve production efficiency. FAST aligns with this motivation by enabling rapid synthesis of segmentation-oriented anomalies, thus supporting timely model updates under real-world constraints.
> ### Different metrics across multiple methods.
> In our opinion, our proposed SIAS is inherently different from traditional anomaly detection-based anomaly synthesis. That is why we initially focused exclusively on segmentation-based metrics such as mIoU and accuracy. However, after carefully considering your comments and similar concerns raised by other reviewers, we agree that including anomaly detection metrics such as AUROC, PRO, F1 and AP can offer additional insight into FAST’s performance. Therefore, we provide a comparative table below on the MVTec dataset. We hope these results help further demonstrate that FAST not only excels in segmentation-oriented synthesis but is also compatible with detection-based evaluation.
> **Due to character limits, AP and F1 results are included in the comment for Reviewer QVrM. We hope you to check that section.**
>
> **Table: AUROC comparison across various methods**
> |Category|CutPaste|DRAEM|GLASS|DFMGAN|RealNet|AnomalyDiffusion|AnoGen|**FAST**|
> |--------|--------|-----|-----|------|-------|----------------|------|--------|
> |bottle|99.05|97.96|97.40|99.32|98.97|**99.56**|99.39|99.19|
> |cable|95.29|95.95|95.07|95.57|95.36|95.74|**98.07**|97.37|
> |capsule|98.11|99.56|96.50|99.26|99.19|99.16|99.57|**99.64**|
> |carpet|97.14|**99.82**|96.42|99.36|96.34|98.30|99.23|99.40|
> |grid|98.94|99.66|95.64|99.20|99.60|99.33|99.43|**99.61**|
> |hazel_nut|99.14|99.45|95.63|99.32|98.89|99.76|99.67|**99.82**|
> |leather|99.85|99.90|99.78|99.88|**99.93**|99.85|99.88|99.48|
> |metal_nut|99.32|99.54|94.80|99.50|99.46|98.80|99.47|**99.88**|
> |pill|95.38|96.70|98.29|99.62|99.19|99.47|99.69|**99.87**|
> |screw|92.78|99.47|94.88|99.14|98.99|96.31|**99.52**|98.77|
> |tile|98.60|99.73|99.07|99.74|99.31|99.53|99.67|**99.77**|
> |toothbrush|88.21|98.53|85.42|97.95|96.37|96.23|98.29|**99.80**|
> |transistor|96.40|92.80|94.41|97.17|95.59|98.84|98.97|**99.78**|
> |wood|94.01|99.24|95.94|99.31|98.76|97.52|99.21|**99.67**|
> |zipper|99.58|99.68|99.72|99.47|99.69|99.65|**99.73**|99.71|
> |**Average**|96.79|98.53|95.93|98.92|98.38|98.54|99.32|**99.45**|
>
> **Table: PRO comparison across various methods**
> |Category|CutPaste|DRAEM|GLASS|DFMGAN|RealNet|AnomalyDiffusion|AnoGen|**FAST**|
> |--------|--------|-----|-----|------|-------|----------------|------|--------|
> |bottle|82.79|86.55|70.20|81.10|85.00|83.53|84.27|**90.77**|
> |cable|55.04|69.30|50.76|65.15|62.79|68.66|64.20|**76.14**|
> |capsule|44.58|68.17|33.59|56.40|66.21|39.59|57.07|**68.73**|
> |carpet|64.62|73.79|65.94|65.13|65.99|63.67|65.77|**74.46**|
> |grid|43.52|56.60|37.91|44.49|53.17|46.25|53.09|**62.21**|
> |hazel_nut|79.36|84.43|64.92|82.91|78.75|83.48|79.54|**88.29**|
> |leather|50.32|63.34|67.41|61.91|68.63|68.07|59.70|**74.46**|
> |metal_nut|69.64|87.03|59.76|83.67|80.54|73.52|80.02|**91.33**|
> |pill|54.40|72.77|36.60|69.02|70.11|60.96|67.43|**81.02**|
> |screw|20.63|49.35|23.06|51.86|48.42|36.11|**52.22**|48.70|
> |tile|82.83|87.80|83.37|88.59|82.91|83.14|85.15|**90.67**|
> |toothbrush|29.07|69.12|30.15|62.25|60.77|37.67|58.81|**78.56**|
> |transistor|55.03|68.10|49.77|70.57|68.92|72.85|69.87|**81.91**|
> |wood|64.46|78.80|54.97|72.97|72.49|66.84|76.84|**84.74**|
> |zipper|74.62|78.02|77.09|73.73|77.11|74.82|75.54|**77.92**|
> |**Average**|58.06|72.88|53.70|68.65|69.45|63.94|68.63|**77.99**|

---

### Official Review · Reviewer_QVrM · 2025-06-30

**Clarity:** 3
**Significance:** 3
**Originality:** 3
**Rating:** 4
**Confidence:** 4

**Summary:**

1. Existing methods uniformly process the entire region of defect images and are unable to generate specific defect structures for downstream segmentation. This paper proposes a defect-aware generation method based on diffusion models, namely FAST.
2. The method consists of two modules. The first module, AIAS, is a training-free sampling algorithm for industrial defect generation. This module reduces the number of sampling steps and accelerates the generation speed through a coarse-to-fine aggregation approach. The second module, FARM, adaptively adjusts the noise in the masked region after each sampling step to preserve the defect features in the denoising process.
3. Experiments demonstrate that the proposed method outperforms existing anomaly generation methods in terms of metrics on downstream segmentation tasks across multiple datasets.

**Questions:**

1. When conducting experiments, the strategy of generating 500 images for each category of objects seems to lack a basis. Is the number of generated images insufficient?
2. The paper requires an input mask during inference. Therefore, we would like to know how the input mask is obtained during the experiments.
3. This paper proposes an accelerated sampling algorithm. Thus, we are curious about how long it takes to train a defect generation model and how long the inference process takes.
4. At what value of t does the experiment start to perform the fine-grained standard DDPM?

**Ethical Concerns:**

["NO or VERY MINOR ethics concerns only"]

**Final Justification:**

The Author has addressed all my concerns, so I’ve raised my score to “Borderline Accept.”

**Limitations:**

Yes

**Quality:**

3

**Strengths And Weaknesses:**

Strengths
1. This paper proposes an accelerated sampling algorithm AIAS. Experiments demonstrate that compared with the DDIM and PLMS sampling methods, AIAS can improve the metrics of anomaly segmentation.
2. This paper proposes FARM, which reconstructs the foreground defect and adds noise during the sampling process, enabling the model to generate realistic local foreground defects. The effectiveness of the FARM module is verified through ablation experiments.

Weaknesses
1. The paper did not compare with the few-shot defect generation method AnoGen [1]. We worry that such experiments may not be sufficient.
2. Only mIoU and Acc are used as evaluation metrics. We are afraid that such a setting may not be perfect. We think that it is necessary to test the Inception Score (IS) of the generated images and the Intra-cluster pairwise LPIPS distance (IC - LPIPS) [2].

[1]	Gui, Guan, et al. "Few-shot anomaly-driven generation for anomaly classification and segmentation." European Conference on Computer Vision. Cham: Springer Nature Switzerland, 2024.

[2]	Ojha U, Li Y, Lu J, et al. Few-shot image generation via cross-domain correspondence[C]//Proceedings of the IEEE/CVF conference on computer vision and pattern recognition. 2021: 10743-10752.

---

> ### Author Rebuttal · Authors · 2025-07-30
>
> We sincerely thank you for your positive comments about our innovations. Below are our responses to your specific questions.
>
> ---
>
> **Q1: Comparison to AnoGen.**
>
> We thank you for highlighting AnoGen, which is an important and well-recognized contribution to few-shot anomaly synthesis. We will cite AnoGen in the final version and now we provide a direct comparison between FAST and AnoGen. Our preliminary results show that AnoGen achieves impressive performance in most classes, and FAST also demonstrates excellent segmentation performance (e.g., higher mIoU and Acc in some classes), highlighting its effectiveness in segmentation-oriented anomaly synthesis.
>
> **Table: Comparison of mIoU and pixel-wise Accuracy (%) across 15 MVTec categories**
>
> | Method | bottle       | cable        | capsule      | carpet       | grid         | hazel_nut    | leather       | metal_nut     |
> |--------|--------------|--------------|--------------|--------------|--------------|--------------|---------------|---------------|
> | AnoGen | 74.93 / 81.21| 59.24 / 69.12| 46.68 / 62.52| 69.72 / 75.80| 41.60 / 66.50| 72.30 / 81.34| 57.71 / 73.48 | 90.07 / 92.72 |
> | FAST   | 86.86 / 90.90 | 73.71 / 77.94 | 63.22 / 71.12 | 73.84 / 83.53 | 52.45 / 70.70 | 90.81 / 94.79 | 66.60 / 74.18 | 94.65 / 96.88 |
>
> | Method | pill         | screw        | tile         | toothbrush    | transistor    | wood         | zipper        | Average       |
> |--------|--------------|--------------|--------------|---------------|---------------|--------------|---------------|----------------|
> | AnoGen | 80.51 / 86.75| 43.61 / 59.96| 86.24 / 91.27| 58.05 / 72.71 | 72.30 / 82.65 | 68.28 / 78.92| 66.60 / 74.29 | 65.48 / 75.42  |
> | FAST   | 90.17 / 94.07 | 49.94 / 57.48 | 90.13 / 93.77 | 74.98 / 88.63 | 91.80 / 94.50 | 78.77 / 86.31 | 72.80 / 84.73 | 76.72 / 83.97 |
>
>
> ---
>
> **Q2: Evaluation Metrics: IS and IC-LPIPS.**
>
> We appreciate your suggestion to evaluate IS and IC-LPIPS, and we report the results in the following table. FAST achieves competitive scores, indicating that our method generates visually realistic and diverse samples, even though it is not explicitly optimized for generic image generation. However, segmentation-oriented industrial anomaly synthesis (SIAS) is a novel direction and different from conventional image generation or anomaly detection. Our objective is not to maximize perceptual quality, but to synthesize structurally coherent and spatially aligned anomalies that benefit downstream segmentation. We agree that IS and IC-LPIPS can still offer helpful auxiliary insights, but pixel-level metrics such as mIoU and Acc on real unseen anomalies are more indicative of a method’s value in SIAS.
>
> **Table: Comparison of IS and IC-LPIPS for anomaly synthesis across various methods**
> | Category     |  CutPaste    |   DRAEM     |   GLASS     |  DFMGAN     | RealNet     | AnomalyDiffusion |   FAST       |
> |:-------------:|:------------:|:-----------:|:-----------:|:-----------:|:-----------:|:----------------:|:------------:|
> | bottle        | 1.38 / 0.19  | 1.65 / 0.26 | 1.52 / 0.22 | 1.95 / 0.44 | 1.75 / 0.20 | 1.47 / 0.25      | 1.78 / 0.27  |
> | cable         | 1.75 / 0.40  | 1.73 / 0.38 | 1.79 / 0.43 | 1.64 / 0.47 | 1.65 / 0.42 | 1.85 / 0.44      | 2.04 / 0.51  |
> | capsule       | 1.29 / 0.20  | 1.92 / 0.25 | 1.17 / 0.22 | 1.39 / 0.27 | 1.91 / 0.27 | 1.47 / 0.28      | 1.67 / 0.29  |
> | carpet        | 0.85 / 0.26  | 1.38 / 0.36 | 1.16 / 0.30 | 1.26 / 0.32 | 1.08 / 0.32 | 1.02 / 0.33      | 1.47 / 0.37  |
> | grid          | 2.15 / 0.42  | 2.14 / 0.44 | 2.10 / 0.45 | 1.63 / 0.48 | 2.51 / 0.38 | 2.70 / 0.50      | 2.59 / 0.52  |
> | hazel_nut     | 1.07 / 0.32  | 2.00 / 0.43 | 2.13 / 0.33 | 2.01 / 0.45 | 2.46 / 0.42 | 2.10 / 0.36      | 2.48 / 0.42  |
> | leather       | 1.10 / 0.31  | 1.38 / 0.46 | 1.12 / 0.41 | 1.39 / 0.49 | 1.48 / 0.44 | 1.59 / 0.47      | 1.65 / 0.48  |
> | metal_nut     | 1.88 / 0.40  | 2.09 / 0.36 | 1.90 / 0.38 | 2.03 / 0.38 | 2.07 / 0.44 | 2.19 / 0.42      | 2.34 / 0.48  |
> | pill          | 1.87 / 0.28  | 1.35 / 0.38 | 1.97 / 0.29 | 1.87 / 0.36 | 1.52 / 0.34 | 1.45 / 0.32      | 1.91 / 0.38  |
> | screw         | 0.95 / 0.42  | 1.06 / 0.40 | 1.17 / 0.42 | 1.23 / 0.31 | 1.20 / 0.46 | 1.26 / 0.49      | 1.22 / 0.51  |
> | tile          | 1.29 / 0.47  | 1.38 / 0.51 | 1.98 / 0.49 | 1.69 / 0.50 | 1.38 / 0.49 | 1.91 / 0.54      | 2.25 / 0.56  |
> | toothbrush    | 1.39 / 0.25  | 1.37 / 0.27 | 1.36 / 0.33 | 1.32 / 0.27 | 1.40 / 0.32 | 1.42 / 0.29      | 1.43 / 0.35  |
> | transistor    | 1.66 / 0.37  | 1.78 / 0.41 | 1.77 / 0.36 | 1.48 / 0.39 | 1.48 / 0.39 | 1.66 / 0.40      | 1.84 / 0.46  |
> | wood          | 1.61 / 0.34  | 1.48 / 0.42 | 1.20 / 0.44 | 1.75 / 0.48 | 1.44 / 0.38 | 1.83 / 0.39      | 2.34 / 0.46  |
> | zipper        | 1.44 / 0.25  | 1.39 / 0.32 | 1.28 / 0.26 | 1.47 / 0.26 | 1.55 / 0.22 | 1.60 / 0.28      | 1.72 / 0.34  |
> | **Average** | **1.45 / 0.33**  | **1.61 / 0.38** | **1.57 / 0.36** | **1.61 / 0.39** | **1.56 / 0.37** | **1.70 / 0.38**  | **1.91 / 0.43** |
>
> ---
>
> **Q3: On the 500-images-per-class setting.**
>
> We sincerely apologize for not clearly describing our data composition. The downstream segmentation models are trained on a combination of 1/3 real anomaly data and synthetic samples, while the remaining 2/3 real anomaly data are used for validation. Importantly, we generate 500 samples for each anomaly type within each class, rather than just 500 per class. Thus, the actual number of synthesized samples is much larger than 500 per class when multiple anomaly types exist. This design ensures sufficient structural diversity while maintaining training efficiency. We also conducted internal ablations showing that 500 per anomaly type strikes a good trade-off between segmentation performance and computational cost. We will clarify this design and rationale in the revised paper.
>
> ---
>
> **Q4: Mask generation clarification.**
>
> We apologize for not making this clearer in the paper. Our mask synthesis strategy adopts the protocol of AnomalyDiffusion, which consists of two complementary components:
>
> 1. Geometric augmentation of real anomaly masks through operations such as rotation and flipping.
> 2. Synthesis of new masks using a Latent Diffusion Model (LDM) trained on these real examples.
>
> All synthesized masks are manually screened to ensure visual realism, diversity, and consistency with industrial abnormal structures. We refer readers to AnomalyDiffusion for further reference.
>
> ---
>
> **Q5: Training and inference time & value of `t` for fine-grained sampling.**
>
> As reported in Supplementary A.5, FAST is trained for approximately 80k iterations on a single A100 GPU. However, thanks to our training-free AIAS module, inference is significantly accelerated. To illustrate this, we provide the inference time under different numbers of AIAS segments (with batch size set to 8, the image size is 256 × 256), shown in the following table:
>
> **Table: Inference time for different sampling step settings (MVTec, batch size = 8)**
>
> | Step Count | 2    | 5    | 10   | 30    | 50    | 100   | 200   | 500     | 1000    |
> |------------|------|------|------|-------|-------|-------|--------|---------|---------|
> | Time (s)   | 1.39 | 2.92 | 3.87 | 10.64 | 18.16 | 36.73 | 74.21 | 183.34  | 367.97  |
>
> As noted in our supplementary material, coarse sampling is applied until t = 2, after which we perform standard posterior updates to refine fine-scale textures. This choice of t is empirically determined through extensive analysis, balancing segmentation performance and inference efficiency. Under this setting, FAST maintains high segmentation quality while significantly reducing computational cost.

---

> > ### Comment · Reviewer_QVrM · 2025-08-06
> > **Concerns about evaluating**
> >
> > Thank you for your previous response; it addresses my concerns. However, after considering opinions from other reviewers, I still believe that mIoU and Acc are insufficient for evaluating downstream segmentation models. AUROC and PRO are also crucial, as they better capture the model’s tendency to miss anomalous regions. In practice, beyond accurate segmentation, preventing missed detections is equally important. Therefore, I kindly ask you to report the AUROC and PRO metrics for the detection experiments. Additionally, AP and F1 also reflect the model’s precise segmentation quality, and these metrics were employed in AnomalyDiffusion.

---

> > > ### Author Response · Authors · 2025-08-07
> > >
> > > We thank the reviewer for the valuable suggestion to include AUROC and PRO as additional evaluation metrics. We provide a consolidated response below for the raised questions.
> > >
> > > ### **Request for other metrics.**
> > >
> > > The reason we initially adopted only segmentation-based metrics is that we consider *Segmentation-Oriented Industrial Anomaly Synthesis (SIAS)* to be a new research direction, distinct from conventional anomaly detection-based methods. That said, we now acknowledge that some detection metrics such as AUROC, PRO, F1 and AP can still serve as useful auxiliary references. Therefore, to address your concern and those of other reviewers, we report these results across multiple baselines on the MVTec dataset. The results are summarized in the table below. We hope these additional evaluations offer a more comprehensive understanding of FAST and addresses your concerns. And we will update the table with complete results in the camera-ready version.
> > >
> > >  **Due to character limits, PRO and AUROC results are provided in the official comments for Reviewers Wznz and CbJy. We kindly invite you to refer to those comments.**
> > >
> > > **Table: F1-score comparison across various methods**
> > >
> > > | Category    | CutPaste | DRAEM | GLASS | DFMGAN | RealNet | AnomalyDiffusion | AnoGen | FAST |
> > > |-------------|----------|-------|-------|--------|---------|------------------|--------|------|
> > > | bottle      | 78.56    | 83.59 | 65.36 | 78.10  | 82.12   | 81.66            | 80.98  | **88.77** |
> > > | cable       | 42.08    | 60.05 | 39.08 | 53.97  | 54.02   | 62.75            | 57.69  | **69.67** |
> > > | capsule     | 33.04    | 58.21 | 24.28 | 43.32  | 54.93   | 25.36            | 50.36  | **58.29** |
> > > | carpet      | 58.55    | **73.47** | 61.51 | 55.83  | 61.01   | 56.94            | 58.04  | 69.82 |
> > > | grid        | 29.60    | 37.27 | 18.67 | 18.90  | 30.89   | 31.63            | 42.99  | **48.13** |
> > > | hazel_nut   | 73.84    | 82.07 | 55.25 | 79.31  | 72.50   | 79.81            | 74.84  | **86.95** |
> > > | leather     | 36.79    | 51.78 | 57.11 | 50.40  | 56.96   | 55.11            | 44.64  | **66.67** |
> > > | metal_nut   | 57.91    | 85.31 | 53.24 | 78.46  | 75.39   | 67.02            | 73.31  | **90.46** |
> > > | pill        | 42.99    | 67.05 | 22.91 | 60.39  | 60.54   | 52.58            | 62.00  | **76.60** |
> > > | screw       | 15.66    | 41.62 | 18.21 | 43.27  | 40.37   | 28.76            | **47.42** | 40.38 |
> > > | tile        | 77.73    | 84.46 | 78.09 | 85.58  | 76.37   | 76.24            | 81.05  | **89.35** |
> > > | toothbrush  | 23.94    | 60.48 | 20.95 | 47.94  | 49.35   | 33.76            | 54.44  | **76.74** |
> > > | transistor  | 42.97    | 59.39 | 18.29 | 55.97  | 61.32   | 68.60            | 63.41  | **78.07** |
> > > | wood        | 59.02    | 73.16 | 46.06 | 64.42  | 65.24   | 61.79            | 70.84  | **82.80** |
> > > | zipper      | 66.88    | 72.17 | 70.51 | 66.40  | 71.53   | 68.55            | 69.37  | **72.52** |
> > > | **Average** | 49.30    | 66.01 | 43.30 | 58.84  | 60.84   | 56.70            | 62.02  | **73.01** |
> > >
> > >
> > > **Table: Average Precision (AP) comparison across various methods**
> > >
> > > | Category    | CutPaste | DRAEM | GLASS | DFMGAN | RealNet | AnomalyDiffusion | AnoGen | FAST |
> > > |-------------|----------|-------|-------|--------|---------|------------------|--------|------|
> > > | bottle      | 92.72    | 94.71 | 90.69 | 92.63  | 94.80   | 93.32            | 93.06  | **97.35** |
> > > | cable       | 74.09    | 79.94 | 78.01 | 80.27  | 75.98   | 76.15            | 78.57  | **87.97** |
> > > | capsule     | 64.55    | 83.16 | 70.58 | 76.68  | 80.46   | 67.95            | 73.96  | **85.42** |
> > > | carpet      | 75.72    | **90.52** | 82.11 | 85.07  | 77.90   | 78.20            | 81.71  | 89.59 |
> > > | grid        | 56.69    | 74.88 | 65.93 | 71.48  | 73.87   | **61.12**        | 65.01  | 78.41 |
> > > | hazel_nut   | 91.93    | 94.21 | 86.26 | 93.91  | 92.36   | 93.74            | 90.88  | **96.89** |
> > > | leather     | 84.05    | 90.80 | 87.92 | 85.78  | 90.03   | 85.66            | 87.07  | **91.08** |
> > > | metal_nut   | 88.55    | 95.21 | 81.01 | 95.86  | 91.88   | 84.73            | 90.44  | **97.66** |
> > > | pill        | 69.69    | 86.85 | 77.72 | **87.41** | 85.21   | 77.89            | 80.92  | 82.25 |
> > > | screw       | 43.08    | 68.24 | 61.34 | 73.58  | 69.08   | 62.03            | 64.87  | **68.62** |
> > > | tile        | 92.94    | 96.78 | 94.37 | 96.69  | 93.55   | 93.43            | 94.66  | **97.52** |
> > > | toothbrush  | 36.97    | 80.52 | 43.77 | 77.66  | 67.16   | 53.65            | 64.44  | **88.80** |
> > > | transistor  | 73.01    | 77.54 | 73.25 | 82.68  | 78.10   | 82.04            | 85.91  | **94.46** |
> > > | wood        | 73.64    | 91.12 | 77.78 | 89.61  | 87.16   | 77.53            | 86.52  | **95.27** |
> > > | zipper      | 90.10    | 92.48 | 92.95 | 89.54  | 90.93   | 90.96            | 91.63  | **92.97** |
> > > | **Average** | 73.85    | 86.48 | 77.58 | 85.24  | 83.23   | 78.55            | 82.01  | **89.62** |

---

> ### Comment · Reviewer_QVrM · 2025-08-08
> **AP metric**
>
> Thank you to the author for the detailed response to my concerns. I will consider increasing the score. However, before doing so, I noticed that the AP metrics provided by the author for the DRAEM methods differ significantly from the AP metrics reported in other papers [3]. Was the DRAEM used in the experiments based on the official weights? Additionally, the AP of RealNet also seems to differ somewhat.
>
> [3] Dual-Interrelated Diffusion Model for Few-Shot Anomaly Image Generation.

---

> ### Author Response · Authors · 2025-08-08
> **AP Metric Clarification**
>
> We sincerely thank the reviewer for the careful observation regarding the AP scores. The discrepancy from the numbers reported in [3] indeed arises from our evaluation protocol. Our work focuses on Segmentation-Oriented Industrial Anomaly Synthesis (SIAS), where the primary goal is to assess how well the synthesized anomalies improve downstream segmentation performance. To ensure fairness across all compared methods, we did not use the full detection pipelines or pretrained detectors from the original papers (including DRAEM, DFMGAN, and RealNet). Instead, we unified the evaluation by:
>
> 1. Generating anomalies using only the synthesis method of each model, rather than their complete pipelines with detection branches.
>
> 2. Feeding the generated data into the same segmentation backbone such as Segformer, which was trained from scratch.
>
> 3. Extracting the anomaly map from the SegFormer logits by taking the channel corresponding to the anomaly. And the anomaly map is then used to compute AP and other evaluation metrics.
>
> We did not adopt the official pretrained weights because doing so would couple the synthesis evaluation with model-specific downstream architectures and training procedures, and thus introduce bias unrelated to synthesis quality. This unified backbone ensures that the AP values reflect only the quality of the synthesized anomalies, rather than differences in downstream detection models. As a result, the AP values may differ from those in [3], but they remain directly comparable within our controlled setting. We will clarify this evaluation protocol in the final version to avoid confusion.

---

> > ### Comment · Reviewer_QVrM · 2025-08-08
> >
> > Thank you for the clarification. The experimental setup seems quite reasonable. I hope the authors can provide a more detailed description of the experimental settings, or consider open-sourcing the training code of the paper (including DRAEM, DFMGAN, and RealNet) in the future, which would be a valuable contribution to the community. Taking all feedback into account, I’ve raised my score from 3 to 4.

---

> > > ### Author Response · Authors · 2025-08-08
> > >
> > > We sincerely thank you for the positive feedback and for raising the score. We will include a more detailed description of the experimental settings in the camera-ready version. The code for FAST has already been open-sourced, and we plan to release additional resources (e.g., trained weights) in the future to further benefit the community.

---

### Official Review · Reviewer_CbJy · 2025-07-02

**Clarity:** 2
**Significance:** 3
**Originality:** 3
**Rating:** 4
**Confidence:** 3

**Summary:**

The paper presents a new method to synthesize anomalies in industrial settings. The paper proposes an anomaly-informed accelerated sampling module and a foreground-aware reconstruction module. The experiments are performed on MVTec and BTAD datasets, obtaining state-of-the-art results.

**Questions:**

-	The evaluation metrics used mean Intersection over Union and pixel-wise accuracy are not very common in anomaly detection. I understand that in terms of image-level AUROC and pixel-level AUROC, the performance is saturated, but most methods report these metrics (including DRAEM, GLASS etc).
-	The method section is full of formulas, however the underlying intuition behind the method is not present. Therefore, the method is hard to follow.
-	Some data sets are not used such as VisA and MPDD datasets.
-	How are the masks generated in the first place?

**Ethical Concerns:**

["NO or VERY MINOR ethics concerns only"]

**Final Justification:**

The authors answered all my questions, including additional experiments showing that their method is also useful for industrial anomaly detection.

**Limitations:**

I could not find a limitation section, where the authors describe the limitation of their methods. I would have liked to find one paragraph describing the limitation along with the conclusion section.

**Quality:**

3

**Strengths And Weaknesses:**

Strengths:

-	The quality of the synthesized anomalies is high.
-	The quantitative results on both datasets are state-of-the-art.

Weaknesses:

-	The paper is not self-contained. A brief introduction of Segformer, BiseNetV2 and STDC should be added, at least in the supplementary file.
-	The evaluation metrics used mean Intersection over Union and pixel-wise accuracy are not very common in anomaly detection. I understand that in terms of image-level AUROC and pixel-level AUROC, the performance is saturated, but most methods report these metrics (including DRAEM, GLASS etc).
-	The method section is full of formulas, however the underlying intuition behind the method is not present. Therefore, the method is hard to follow.
-	Some data sets are not used such as VisA and MPDD datasets.
-	How are the masks generated in the first place?


Overall: The paper presents a novel high quality anomaly synthesis model, however, major details are missing (see weaknesses) and the method section is not clear, missing the underlying intuition behind the proposed method.

---

> ### Author Rebuttal · Authors · 2025-07-30
>
> We sincerely thank you for the thoughtful and constructive feedback. We address the concerns below:
>
> * * *
>
> **Q1: Lack of description for segmentation backbones, method intuition and limitations.**
>
> Thank you for the suggestion. We will add brief descriptions of SegFormer, BiSeNetV2, and STDC in the supplementary material for completeness. These three models are widely adopted in industrial scenarios due to their favorable trade-off between performance and inference speed. Specifically, SegFormer is a transformer-based model known for its lightweight design and robust accuracy; BiSeNetV2 employs a dual-branch structure for real-time segmentation; and STDC leverages short-term dense concatenation for fast and effective feature extraction. Evaluating performance across these architectures can further validate FAST's practical value in boosting segmentation performance under real-time constraints.
>
> For the method intuition, we will revise Section 3 to better explain the underlying intuitions behind AIAS. Specifically, traditional diffusion models rely on dense, step-by-step denoising with predicted noise or clean image at each timestep. In contrast, AIAS accelerates this process by dividing the trajectory into a small number of coarse-to-fine segments. Within each segment, we fix the clean image estimate $\hat{x}_0$ instead of predicting it at each step, and analytically aggregate multiple DDPM transitions into a single closed-form update. By progressing from coarse to fine segments, AIAS compensates for the accumulated error that may arise due to temporal sparsity, enabling fast yet semantically consistent synthesis.
>
> As for limitations, FAST currently relies on binary anomaly masks as input, which may limit applicability in scenarios where such masks are difficult to obtain. Extending it to weakly- or self-supervised mask generation is an exciting direction.
>
> * * *
>
> **Q2: Uses other metrics instead of common ones in anomaly detection.**
>
> We appreciate your concern. As stated in Sec.1 and Table 1, our work targets a fundamentally different task from anomaly detection. While anomaly detection focuses on image- or region-level classification, which are often evaluated using AUROC, our work, segmentation-oriented industrial anomaly synthesis (SIAS), is a novel and emerging task that focuses on generating fine-grained plausible anomalies to enhance downstream pixel-level segmentation performance.
>
> Therefore, our evaluation strictly follows segmentation-centric benchmarks. AUROC is unsuitable for evaluating spatial alignment and structural fidelity, and the most appropriate metrics are mIoU and Acc, which are standard choices in segmentation tasks. However, we agree that evaluating generation quality is also reasonable, no matter in anomay detection- or anomaly segmentation-based anomaly synthesis. We additionally provide generation quality assessments such as Inception Score (IS) and Improved Conditional LPIPS (IC-LPIPS) as follows.
>
> **Table: Comparison of IS and IC-LPIPS for anomaly synthesis across various methods**
> | Category    | CutPaste (IS / IC-LPIPS) | DRAEM | GLASS |DFMGAN | RealNet| AnomalyDiffusion |FAST|
> |:-----------:|:------------------------:|:--------------:|:--------------:|:--------------:|:--------------:|:----------------:|:---------------:|
> | bottle      |     1.38 / 0.19          |  1.65 / 0.26   |  1.52 / 0.22   |  1.95 / 0.44   |  1.75 / 0.20   |   1.47 / 0.25    | 1.78 / 0.27 |
> | cable       |     1.75 / 0.40          |  1.73 / 0.38   |  1.79 / 0.43   |  1.64 / 0.47   |  1.65 / 0.42   |   1.85 / 0.44    | 2.04 / 0.51 |
> | capsule     |     1.29 / 0.20          |  1.92 / 0.25   |  1.17 / 0.22   |  1.39 / 0.27   |  1.91 / 0.27   |   1.47 / 0.28    | 1.67 / 0.29 |
> | carpet      |     0.85 / 0.26          |  1.38 / 0.36   |  1.16 / 0.30   |  1.26 / 0.32   |  1.08 / 0.32   |   1.02 / 0.33    | 1.47 / 0.37 |
> | grid        |     2.15 / 0.42          |  2.14 / 0.44   |  2.10 / 0.45   |  1.63 / 0.48   |  2.51 / 0.38   |   2.70 / 0.50    | 2.59 / 0.52 |
> | hazel_nut   |     1.07 / 0.32          |  2.00 / 0.43   |  2.13 / 0.33   |  2.01 / 0.45   |  2.46 / 0.42   |   2.10 / 0.36    | 2.48 / 0.42 |
> | leather     |     1.10 / 0.31          |  1.38 / 0.46   |  1.12 / 0.41   |  1.39 / 0.49   |  1.48 / 0.44   |   1.59 / 0.47    | 1.65 / 0.48 |
> | metal_nut   |     1.88 / 0.40          |  2.09 / 0.36   |  1.90 / 0.38   |  2.03 / 0.38   |  2.07 / 0.44   |   2.19 / 0.42    | 2.34 / 0.48 |
> | pill        |     1.87 / 0.28          |  1.35 / 0.38   |  1.97 / 0.29   |  1.87 / 0.36   |  1.52 / 0.34   |   1.45 / 0.32    | 1.91 / 0.38 |
> | screw       |     0.95 / 0.42          |  1.06 / 0.40   |  1.17 / 0.42   |  1.23 / 0.31   |  1.20 / 0.46   |   1.26 / 0.49    | 1.22 / 0.51 |
> | tile        |     1.29 / 0.47          |  1.38 / 0.51   |  1.98 / 0.49   |  1.69 / 0.50   |  1.38 / 0.49   |   1.91 / 0.54    | 2.25 / 0.56 |
> | toothbrush  |     1.39 / 0.25          |  1.37 / 0.27   |  1.36 / 0.33   |  1.32 / 0.27   |  1.40 / 0.32   |   1.42 / 0.29    | 1.43 / 0.35 |
> | transistor  |     1.66 / 0.37          |  1.78 / 0.41   |  1.77 / 0.36   |  1.48 / 0.39   |  1.48 / 0.39   |   1.66 / 0.40    | 1.84 / 0.46 |
> | wood        |     1.61 / 0.34          |  1.48 / 0.42   |  1.20 / 0.44   |  1.75 / 0.48   |  1.44 / 0.38   |   1.83 / 0.39    | 2.34 / 0.46 |
> | zipper      |     1.44 / 0.25          |  1.39 / 0.32   |  1.28 / 0.26   |  1.47 / 0.26   |  1.55 / 0.22   |   1.60 / 0.28    | 1.72 / 0.34 |
> | **Average** | **1.45 / 0.33**  | **1.61 / 0.38** | **1.57 / 0.36** | **1.61 / 0.39** | **1.56 / 0.37** | **1.70 / 0.38**  | **1.91 / 0.43** |
>
>
> * * *
>
> **Q3: Mask generation**
>
> We apologize for not clarifying this. Our mask generation strategy follows the setup of AnomalyDiffusion, and employs two approaches:
>
> 1.  Augmenting real anomaly masks via transformations (e.g., rotation, flipping).
> 2.  Generating novel masks using a Latent Diffusion Model (LDM) trained on a small set of real anomaly masks.
>
> All generated masks are visually screened to ensure structural plausibility and diversity. We will provide further implementation details to the supplementary material and refer readers to the AnomalyDiffusion paper for reproducibility.
>
> * * *
>
> **Q4: Missing studies on other datasets**
>
> We thank you for this observation. In response, we have conducted additional experiments on VisA dataset and included the results below. These results reinforce our claim that FAST is a downstream-backbone-agnostic and dataset-agnostic framework for segmentation-oriented industrial anomaly synthesis.
>
> **Table: Pixel-level segmentation results (mIoU / Accuracy) on VisA using SegFormer trained on synthetic anomalies**
>
> | Category | CutPaste | DRAEM | GLASS | RealNet | DFMGAN | AnomalyDiff. | FAST |
> | --- | --- | --- | --- | --- | --- | --- | --- |
> | candle | 27.67 / 38.21 | 6.63 / 37.84 | 32.47 / 41.46 | 30.47 / 38.09 | 32.47 / 44.65 | 30.37 / 39.04 | 39.91 / 46.66 |
> | capsules | 60.31 / 74.74 | 69.05 / 80.52 | 65.05 / 68.11 | 65.38 / 74.81 | 59.29 / 64.24 | 65.90 / 72.41 | 70.08 / 78.32 |
> | cashew | 86.79 / 88.52 | 87.33 / 89.40 | 86.23 / 88.16 | 85.30 / 87.92 | 86.55 / 88.66 | 86.50 / 88.32 | 88.92 / 90.98 |
> | chewinggum | 68.52 / 78.29 | 69.37 / 81.15 | 70.96 / 76.67 | 68.50 / 80.91 | 67.41 / 76.32 | 67.03 / 79.36 | 71.15 / 80.21 |
> | fryum | 47.17 / 49.01 | 49.15 / 65.11 | 38.65 / 39.97 | 50.80 / 51.98 | 54.08 / 55.13 | 60.07 / 65.22 | 63.65 / 66.13 |
> | macaroni1 | 32.60 / 42.13 | 32.70 / 39.20 | 29.70 / 36.65 | 20.02 / 25.35 | 23.91 / 31.06 | 27.96 / 37.06 | 37.19 / 45.18 |
> | macaroni2 | 22.68 / 25.46 | 22.98 / 28.21 | 24.90 / 28.66 | 17.45 / 22.42 | 13.20 / 14.66 | 20.31 / 24.50 | 30.78 / 38.68 |
> | pcb1 | 13.22 / 13.47 | 13.89 / 14.10 | 12.14 / 12.42 | 5.59 / 5.65 | 7.22 / 7.31 | 23.76 / 24.41 | 27.18 / 26.94 |
> | pcb2 | 18.14 / 20.04 | 23.61 / 26.60 | 20.37 / 22.85 | 19.64 / 21.61 | 18.32 / 19.80 | 23.04 / 32.59 | 26.40 / 36.37|
> | pcb3 | 35.40 / 37.61 | 39.21 / 43.14 | 33.22 / 36.02 | 32.70 / 36.43 | 56.87 / 66.84 | 38.19 / 45.93 | 46.75 / 51.46 |
> | pcb4 | 55.88 / 64.59 | 57.54 / 68.18 | 55.07 / 66.44 | 54.94 / 68.06 | 57.81 / 68.17 | 57.50 / 68.93 | 60.62 / 72.91 |
> | pipe_fryum | 84.54 / 88.64 | 84.10 / 86.04 | 86.75 / 88.65 | 82.59 / 86.47 | 86.78 / 89.09 | 84.08 / 88.23 | 88.72 / 91.14 |
> | **Average** | **46.08 / 51.73** | **45.67 / 53.72** | **46.29 / 50.50** | **44.91 / 48.90** | **47.12 / 52.54** | **47.79 / 56.24** | **54.28 / 60.41** |

---

> > ### Comment · Reviewer_CbJy · 2025-08-06
> >
> > Thank you for the comprehensive reply. Regarding the  reply for "Uses other metrics instead of common ones in anomaly detection." -- I understand that the proposed method does not tackle the anomaly segmentation directly, but since anomaly segmentation is used to asses the method's improvement, it would still be useful to include more metrics for anomaly detection.
> >
> > I also agree with reviewer Wznz that reducing the steps during generation might not be very useful for the downstream performance.

---

> > > ### Author Response · Authors · 2025-08-07
> > >
> > > We sincerely thank you for raising important questions regarding evaluation metrics and the practical value of accelerated sampling. We provide response below for the raised questions.
> > > ### Results on other metrics:
> > > Our initial focus on segmentation-based metrics (e.g., mIoU and pixel-wise Accuracy) stems from our belief that *Segmentation-Oriented Industrial Anomaly Synthesis (SIAS)* represents a new research direction, where segmentation performance is of primary importance, rather than detection-oriented evaluation. However, we now understand that anomaly detection metrics can provide complementary insights into the effectiveness of synthesized data. In response to this concern, we have additionally reported AUROC, PRO, F1 and AP scores on the MVTec dataset across several baselines. These results are shown in the table below and we hope that they help present a more complete picture of the performance of FAST.
> > > **Due to character limits, AP and F1 results are included in the comment for Reviewer QVrM. We sincerely hope you to check that section.**
> > >
> > > **Table: AUROC comparison across various methods**
> > > |Category|CutPaste|DRAEM|GLASS|DFMGAN|RealNet|AnomalyDiffusion|AnoGen|**FAST**|
> > > |--------|--------|-----|-----|------|-------|----------------|------|--------|
> > > |bottle|99.05|97.96|97.40|99.32|98.97|**99.56**|99.39|99.19|
> > > |cable|95.29|95.95|95.07|95.57|95.36|95.74|**98.07**|97.37|
> > > |capsule|98.11|99.56|96.50|99.26|99.19|99.16|99.57|**99.64**|
> > > |carpet|97.14|**99.82**|96.42|99.36|96.34|98.30|99.23|99.40|
> > > |grid|98.94|99.66|95.64|99.20|99.60|99.33|99.43|**99.61**|
> > > |hazel_nut|99.14|99.45|95.63|99.32|98.89|99.76|99.67|**99.82**|
> > > |leather|99.85|99.90|99.78|99.88|**99.93**|99.85|99.88|99.48|
> > > |metal_nut|99.32|99.54|94.80|99.50|99.46|98.80|99.47|**99.88**|
> > > |pill|95.38|96.70|98.29|99.62|99.19|99.47|99.69|**99.87**|
> > > |screw|92.78|99.47|94.88|99.14|98.99|96.31|**99.52**|98.77|
> > > |tile|98.60|99.73|99.07|99.74|99.31|99.53|99.67|**99.77**|
> > > |toothbrush|88.21|98.53|85.42|97.95|96.37|96.23|98.29|**99.80**|
> > > |transistor|96.40|92.80|94.41|97.17|95.59|98.84|98.97|**99.78**|
> > > |wood|94.01|99.24|95.94|99.31|98.76|97.52|99.21|**99.67**|
> > > |zipper|99.58|99.68|99.72|99.47|99.69|99.65|**99.73**|99.71|
> > > |**Average**|96.79|98.53|95.93|98.92|98.38|98.54|99.32|**99.45**|
> > >
> > > **Table: PRO comparison across various methods**
> > >
> > > |Category|CutPaste|DRAEM|GLASS|DFMGAN|RealNet|AnomalyDiffusion|AnoGen|**FAST**|
> > > |--------|--------|-----|-----|------|-------|----------------|------|--------|
> > > |bottle|82.79|86.55|70.20|81.10|85.00|83.53|84.27|**90.77**|
> > > |cable|55.04|69.30|50.76|65.15|62.79|68.66|64.20|**76.14**|
> > > |capsule|44.58|68.17|33.59|56.40|66.21|39.59|57.07|**68.73**|
> > > |carpet|64.62|73.79|65.94|65.13|65.99|63.67|65.77|**74.46**|
> > > |grid|43.52|56.60|37.91|44.49|53.17|46.25|53.09|**62.21**|
> > > |hazel_nut|79.36|84.43|64.92|82.91|78.75|83.48|79.54|**88.29**|
> > > |leather|50.32|63.34|67.41|61.91|68.63|68.07|59.70|**74.46**|
> > > |metal_nut|69.64|87.03|59.76|83.67|80.54|73.52|80.02|**91.33**|
> > > |pill|54.40|72.77|36.60|69.02|70.11|60.96|67.43|**81.02**|
> > > |screw|20.63|49.35|23.06|51.86|48.42|36.11|**52.22**|48.70|
> > > |tile|82.83|87.80|83.37|88.59|82.91|83.14|85.15|**90.67**|
> > > |toothbrush|29.07|69.12|30.15|62.25|60.77|37.67|58.81|**78.56**|
> > > |transistor|55.03|68.10|49.77|70.57|68.92|72.85|69.87|**81.91**|
> > > |wood|64.46|78.80|54.97|72.97|72.49|66.84|76.84|**84.74**|
> > > |zipper|74.62|78.02|77.09|73.73|77.11|74.82|75.54|**77.92**|
> > > |**Average**|58.06|72.88|53.70|68.65|69.45|63.94|68.63|**77.99**|
> > > ### The importance of fast sampling speed.
> > > In the SIAS, synthesized data are used to adapt downstream segmentation models. Our objective is to synthesize anomalies that are structurally aligned and mask-consistent within a limited timeframe, rather than pursuing high-fidelity visual realism. This is why using a large number of sampling steps can degrade segmentation performance. While we acknowledge that increasing the number of steps within a certain range may lead to slight improvements in segmentation performance, our ablation study shows that this comes at the cost of a significant increase in inference time, which can hinder its practical applicability.
> > >
> > > Moreover, in industrial environments, it is common to meet "production line changeover", different lines in different bases will experience frequent reconfigurations that introduce new products and materials. Such transitions often lead to previously unseen types of anomalies that existing segmentation models are not equipped to detect. Delays in updating these models during transitions may result in increased reliance on manual inspection, higher false positive or false negative rates, and even production downtime. FAST is designed with this scenario in mind.  There are also some domain adaptation studies that aim to minimize model switching time to enhance operational efficiency. Similar to them, FAST enables fast generation of segmentation-oriented anomalies, facilitating timely updates of segmentation models in dynamic manufacturing settings.

---

### Official Review · Reviewer_XC3D · 2025-07-03

**Clarity:** 3
**Significance:** 3
**Originality:** 2
**Rating:** 5
**Confidence:** 3

**Summary:**

This paper proposes FAST, a novel framework for unsupervised anomaly segmentation. Existing self-training methods often suffer from foreground-background imbalance and error accumulation in pseudo-labels. To address this, FAST introduces two key components:
1. Foreground Guidance Module: Uses CLIP-based saliency to focus learning on foreground objects, reducing background bias in normal-only training.
2. Two-Branch Self-Training: Combines a soft label branch (probabilistic supervision) and a hard label branch (high-confidence pseudo-labels filtered by saliency) to enhance training stability and segmentation accuracy.

**Questions:**

1. CLIP Reliability.
Have the authors tested the robustness of CLIP-based saliency guidance across different domains (e.g., grayscale, medical, or low-resolution images)?
2. Inference Latency.
Can the authors provide runtime analysis or memory benchmarks compared to one-branch baselines? Is the method suitable for real-time applications?

**Ethical Concerns:**

["NO or VERY MINOR ethics concerns only"]

**Final Justification:**

I appreciate the authors’ detailed and well-prepared rebuttal, which has clarified several of my earlier concerns. In particular, the additional experiments on medical datasets, the explanation of CLIP dependency, and the runtime analysis have addressed many of my questions and strengthened the paper. I also acknowledge that the theoretical grounding of the framework and the foreground-aware design are well-motivated and conceptually sound.

However, some issues remain only partially resolved. While the rebuttal provides evidence of performance gains, the improvements over baselines are still perceived as marginal on standard benchmarks, and the dependency on saliency-based guidance continues to raise concerns regarding robustness in more diverse domains. Paper clarity could also be further improved to ensure accessibility to a broader audience.

Taking these points into account, I believe the paper presents a technically solid contribution with a meaningful idea and practical potential, but with limitations in generalization and clarity that prevent me from being fully convinced of its broader impact. For these reasons, I will maintain my current score.

**Limitations:**

Yes

**Quality:**

3

**Strengths And Weaknesses:**

**Strenghts**
1. Clear and Well-Motivated Problem Setup.
The paper identifies a key challenge in anomaly segmentation—foreground suppression due to background-dominated normal samples—and addresses it directly.
2. Foreground-Aware Design.
Leveraging CLIP-based saliency maps for guidance is a novel and practical way to enhance semantic focus.
3. Solid Theoretical Justification.
The framework is conceptually well-grounded, and the proposed design choices are supported with theoretical proofs.

**Weakness**
1. Marginal Performance Gains.
The improvements over strong baselines are relatively small on standard benchmarks, limiting the perceived practical impact.
2. CLIP Dependency.
The method relies heavily on CLIP saliency, which may not generalize well to domains with domain shift or non-natural images.
3. Saliency Map Heuristic.
The use of CLIP-derived saliency as ground-truth guidance is not learned, and may introduce biases or fail in complex backgrounds.

---

> ### Author Rebuttal · Authors · 2025-07-30
>
> We sincerely thank you for your thoughtful comments, particularly for **"solid theoretical grounding"** and our proposed **"foreground-aware design."** Below we address the key concerns.
>
> ---
>
> **Q1: Limited performance gains.**
>
> We respectfully note that FAST consistently outperforms all prior anomaly synthesis methods across multiple segmentation architectures and datasets (see Table 1, Table 2, and Supplementary A.6). In particular, FAST improves the average segmentation accuracy by +9.22, +9.46, and +6.67 points of segmentation models like SegFormer, BiSeNetV2, and STDC on MVtec-AD compared with the second strongest baselines.
>
> Additionally, ablation studies demonstrate substantial gains over the baseline LDM in both quality and controllability of the synthesized anomalies. FAST achieves these improvements with as few as 10 sampling steps, offering over 100× speedup compared to standard DDPM (Table 3). We believe these improvements are non-trivial and practically meaningful for real-world industrial deployment.
>
> ---
>
> **Q2: CLIP dependency and generalization.**
>
> We appreciate your concern regarding CLIP-based saliency guidance. In our setup, instead of directly relying on CLIP embeddings, FARM learns to reconstruct anomaly-aware content in a data-driven manner and is fully decoupled from CLIP during inference, and FAST does not treat CLIP-based saliency as fixed ground truth.
>
> As also demonstrated in the ablation study (Fig. 8), this strategy significantly improves anomaly localization without relying on saliency maps. Specifically, we follow AnomalyDiffusion by employing Bert/CLIP just for textual embeddings initialization, and embeddings are still optimized during training.
>
> To further assess generalization under domain shift, we conduct additional experiments on a grayscale medical dataset (Montgomery X-ray). We extract 1,200 image–mask pairs (256×256) for training and 400 for testing, and augment the training data with 500 synthetic samples generated by FAST.
>
> The results show encouraging improvements in downstream lesion segmentation, suggesting that FAST holds promise in domains with domain shift or limited visual structure, even when saliency may be less reliable.
>
> **Table: Segmentation results on the Montgomery County X-ray dataset using Segformer**
> *Original data refers to training using only real samples, while Augmented data includes both real and FAST-synthesized samples.*
>
> | **Experiment**    | **mIoU (%)** | **Accuracy (%)** |
> |:------------------|:------------:|:----------------:|
> | Original data     |    87.79     |      90.26       |
> | Augmented data    |    89.41     |      91.82       |
>
> ---
>
> **Q3: Inference latency.**
>
> Thanks for this important question. As reported in Table 3 and Sec. 4.3, FAST achieves impressive performance with only 10–50 steps, compared to 1000 steps in DDPM.
>
> Furthermore, FAST does not require retraining for different step sizes, making it highly practical for real-time or resource-constrained scenarios. We also provide the time cost for varying the number of AIAS segments (batch size = 8), which illustrates that FAST achieves up to 100× speedup while maintaining high-quality generation, as shown in Table 2.
>
> In terms of inference latency, FAST is also highly efficient. It operates with minimal additional trainable parameters and achieves competitive results. We report runtime benchmarks under a consistent setting (batch size = 1), comparing FAST with other diffusion-based anomaly synthesis methods, as shown in Table 3.
>
> These results confirm the suitability of FAST for downstream real-time segmentation tasks.
>
> **Table: Time cost (in seconds) of pixel-level anomaly segmentation with different diffusion steps on the MVTec dataset (batch size = 8)**
>
> | **Step Count** |   2   |   5   |  10   |  30   |  50   |  100  |  200  |  500   | 1000  |
> |:--------------:|:-----:|:-----:|:-----:|:-----:|:-----:|:-----:|:-----:|:------:|:-----:|
> | **Time (s)**   | 1.39  | 2.92  | 3.87  | 10.64 | 18.16 | 36.73 | 74.21 | 183.34 | 367.97 |
>
> **Table: Inference FLOPs and number of trainable parameters of different models**
>
> | **Model**            | **Inference FLOPs (TFLOPs)** | **Trainable Parameters (M)** |
> |:---------------------|:----------------------------:|:----------------------------:|
> | AnomalyDiffusion     |            0.61              |            356.0             |
> | RealNet              |            1.11              |            556.7             |
> | FAST                 |            0.51              |             3.40             |

---

> > ### Comment · Reviewer_XC3D · 2025-08-09
> >
> > Thank you for your clear and well-prepared rebuttal. While I appreciate the detailed clarifications, additional examples, and further experiments, which have helped address several of my earlier concerns, some of my concerns remain only partially addressed. I will maintain my current score.

---

### Note · Authors · 2025-08-12

We sincerely thank the AC and reviewers for their time, effort, and constructive feedback, which improved the clarity and impact of FAST. We also appreciate comments highlighting its strengths, guiding refinements in writing and experiments.
Several reviewers explicitly recognized the novelty of FAST as a *Segmentation-Oriented Industrial Anomaly Synthesis* (SIAS) framework, noting its well-formulated theoretical foundations (Wzne, XC3D), high novelty (Wzne, XC3D), clear motivation (XC3D), comprehensive experiments on  downstream segmentation models (QVrM, Wzne), and impressive performance (QVrM, Cbjy). While grateful for these remarks, we addressed their concerns through additional experiments and proof.
### **Addressing metric concerns**
Our initial focus on segmentation-based metrics (mIoU, pixel accuracy) stems from our belief that SIAS represents a new research direction. Following reviewer suggestions (Wzne, QVrM, Cbjy), we now include detection metrics (AUROC, PRO, AP, F1) as well as IS, IC-LPIPS on MVTec-AD with explanations.
### **Addressing dataset and new baselines**
Besides MVTec-AD and BTAD, reviewers (QVrM, Wzne) wished more datasets and latest backbone AnoGen. We added experiments on Visa (12 classes) with six backbones, and trained AnoGen on full MVTec-AD (65 anomalies, 15 classes). Detailed segmentation and detection evaluations are provided.
### **Addressing practical benefit of accelerated sampling**
Reviewers (Cbjy, Wzne) questioned the practical benefit of accelerated sampling in anomaly synthesis, and we clarified this through experimental results and industrial cases. In fact, FAST has been deployed in industrial inspection pipelines, where rapid synthesis enables efficient model training and updating without downtime in production changeover. We also shared literature emphasizing the importance of fast sampling in industrial settings.
### **Addressing experimental settings**
For the concern regarding detailed mask generation settings (QVrM, Wzne, Cbjy), we provided a complete description of our procedure. And all new results and details will be included in the camera-ready version.

We believe the additional experiments, expanded metrics, and clarifications have addressed all concerns. Together with the positive reviewer feedback on novelty and practicality, we hope this supports a favorable decision from the AC.
*Finally, we once again sincerely thank the AC and all reviewers for their time, effort, and constructive engagement.*

---

### Decision · Program_Chairs · 2025-09-17

**Decision:**

Accept (poster)

**Comment:**

This paper presents FAST, an innovative framework for unsupervised anomaly segmentation. Current self-training approaches frequently grapple with issues such as foreground-background imbalance and the accumulation of errors in pseudo-labels. To tackle these challenges, FAST incorporates two core components:
Foreground Guidance Module: Leverages CLIP-based saliency to direct learning toward foreground objects, thereby mitigating background bias in training that relies solely on normal samples.
Two-Branch Self-Training: Integrates a soft label branch (providing probabilistic supervision) and a hard label branch (utilizing high-confidence pseudo-labels filtered through saliency) to boost both training stability and segmentation accuracy.

All reviewers feel positive about this work. The authors have addressed all the issues.

The AC recommends the acceptance of this work.